# Closed-Form Training Dynamics Reveal Learned Features and Linear Structure in `Word2Vec`-like Models

**Dhruva Karkada**[*]
UC Berkeley

**James B. Simon**
Imbue and UC Berkeley

**Yasaman Bahri**
Google DeepMind

**Michael R. DeWeese**
UC Berkeley

## Abstract

Self-supervised word embedding algorithms such as `word2vec` provide a minimal setting for studying representation learning in language modeling. We examine the quartic Taylor approximation of the `word2vec` loss around the origin, and we show that both the resulting training dynamics and the final performance on downstream tasks are empirically very similar to those of `word2vec`. Our main contribution is to analytically solve for both the gradient flow training dynamics and the final word embeddings in terms of only the corpus statistics and training hyperparameters. The solutions reveal that these models learn orthogonal linear subspaces one at a time, each one incrementing the effective rank of the embeddings until model capacity is saturated. Training on Wikipedia, we find that each of the top linear subspaces represents an interpretable topic-level concept. Finally, we apply our theory to describe how linear representations of more abstract semantic concepts emerge during training; these can be used to complete analogies via vector addition.

## 1 Introduction

Modern machine learning models achieve impressive performance on complex tasks in large part due to their ability to automatically extract useful features from data. Despite rapid strides in engineering, a scientific theory describing this process remains elusive. The challenges in developing such a theory include the complexity of model architectures, the nonconvexity of the optimization, and the difficulty of data characterization. To make progress, it is prudent to turn to simple models that admit theoretical analysis while still capturing phenomena of interest.

Word embedding algorithms are a class of self-supervised algorithms that learn word representations with task-relevant vector structure. One example is `word2vec`, a contrastive algorithm that learns to model the probability of finding two given words co-occurring in natural text using a two-layer linear network (Mikolov et al., 2013). Despite its simplicity, the resulting models succeed on a variety of semantic understanding tasks. One striking ability exhibited by these embeddings is analogy completion: most famously, $\mathbf{man} - \mathbf{woman} \approx \mathbf{king} - \mathbf{queen}$, where $\mathbf{man}$ is the embedding for the word "man" and so on. Importantly, this ability is not explicitly promoted by the optimization objective; instead, it emerges spontaneously from the process of representation learning.

**Contributions.** In this work, we give a closed-form description of representation learning in models trained to minimize the quartic Maclaurin approximation of the `word2vec` loss. We prove that the learning problem reduces to matrix factorization with quadratic loss (Theorem 1), so we call these models *quadratic word embedding models* (QWEMs). We derive analytic solutions for their training dynamics and final embeddings under gradient flow and vanishing initialization (Result 3). Training on Wikipedia, we show that QWEMs closely match `word2vec`-trained models in their training dynamics, learned features, and performance on standard benchmarks (Figure 3). We apply our results to show that the dynamical formation of abstract linear representations is well-described by quantities taken from random matrix theory (Figure 4). Taken together, our results give a clear picture of the learning dynamics in contrastive word embedding algorithms such as `word2vec`.

---

[*]dkarkada@berkeley.edu
Code to reproduce all experiments available at https://github.com/dkarkada/qwem.

39th Conference on Neural Information Processing Systems (NeurIPS 2025).

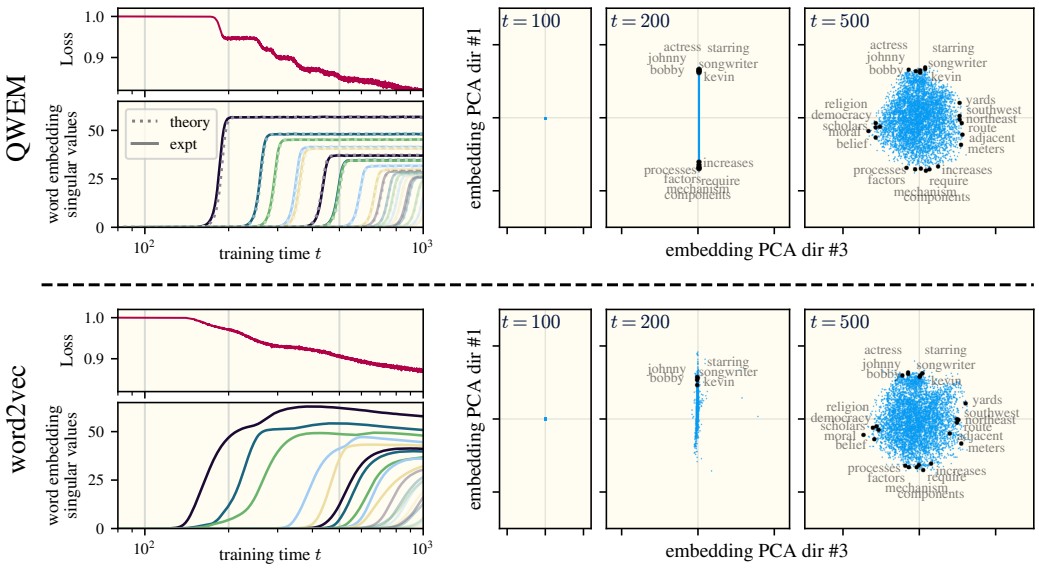

**Figure 1: Quadratic word embedding models are a faithful and analytically solvable proxy for `word2vec`.** We compare the time course of learning in QWEMs (top) and `word2vec` (bottom), finding striking similarities in their training dynamics and learned representations. Analytically, we solve for the optimization dynamics of QWEMs under gradient flow from small initialization, revealing discrete, rank-incrementing learning steps corresponding to stepwise decreases in the loss (top left). In latent space (right side plots), embedding vectors expand into subspaces of increasing dimension at each learning step. These PCA directions are the model's learned features, and they can be extracted from our theory in *closed form* given only the corpus statistics and hyperparameters (Theorem 1). Empirically, QWEMs yield high-quality embeddings very similar to `word2vec`'s in terms of their learned features and performance on benchmarks (Figure 3). See Appendix A for details.

**Relation to previous work.** `word2vec` is a self-supervised contrastive word embedding algorithm widely used for its simplicity and performance (Mikolov et al., 2013; Levy et al., 2015). Although the resulting models are known to implicitly factorize a target matrix (Levy & Goldberg, 2014), it is not known *which* low-rank approximation of the target is learned (Arora et al., 2016). We provide the answer in a close approximation of the task, solving for the final word embeddings directly in terms of the statistics of the data and the training hyperparameters.

Our result connects deeply with previous works on the gradient descent dynamics of linear models. For two-layer linear feedforward networks trained on a supervised learning task with square loss, whitened inputs, and weights initialized to be aligned with the target, the singular values of the weights undergo sigmoidal dynamics; each singular direction is learned independently with a distinct learning timescale (Saxe et al., 2014, 2019; Gidel et al., 2019; Atanasov et al., 2022). These results either rely on assumptions on the data (e.g., input covariance $\Sigma_{xx} = I$ or $\Sigma_{xx}$ commutes with $\Sigma_{xy}$) or are restricted to scalar outputs. Similarly, supervised matrix factorization models are known to exhibit rank-incremental training dynamics in some settings (Li et al., 2018; Arora et al., 2019; Gissin et al., 2019; Li et al., 2021; Jacot et al., 2021; Jiang et al., 2023; Chou et al., 2024). Many of these results rely on over-parameterization or special initialization schemes. While our result is consistent with these works, our derivation does not require assumptions on the data distribution, nor does it require special structure in the initial weights. Another key difference is that our result is the first to solve for the training dynamics of a natural language task learned by a *self-supervised* contrastive algorithm. This directly expands the scope of matrix factorization theory to new settings of interest.

Closest to our work, HaoChen et al. (2021); Tian et al. (2021); Simon et al. (2023b) study linearized contrastive learning in vision tasks. Our work differs in several ways: we study a natural language task using a different contrastive loss function, we do not linearize a nonlinear model architecture, we obtain closed-form solutions for the learning dynamics, and we do not require assumptions on the data distribution (e.g., special graph structure in the image augmentations, or isotropic image data). Saunshi et al. (2022) stress that a theory of contrastive learning must account for both the true data distribution and the optimization dynamics; to our knowledge, our result is the first to do so.

## 2 Preliminaries

**Notation.** We use capital boldface to denote matrices and lowercase boldface for vectors. Subscripts denote elements of vectors and tensors ($\boldsymbol{A}_{ij}$ is a scalar). We use the "economy-sized" singular value decomposition (SVD) $\boldsymbol{A} = \boldsymbol{U}\boldsymbol{S}\boldsymbol{V}^\top$, where $\boldsymbol{S}$ is square. We denote the rank-$r$ truncated SVD as $\boldsymbol{A}_{[r]} = \boldsymbol{U}_{[:,:r]}\boldsymbol{S}_{[:r,:r]}\boldsymbol{V}_{[:,:r]}^\top$.

**Setup.** The training corpus is a long sequence of words drawn from a finite vocabulary of cardinality $V$. A *context* is any contiguous length-$L$ subsequence of the corpus. Let $i$ and $j$ index the vocabulary. Let $\Pr(i)$ be the empirical unigram distribution, and let $\Pr(j|i)$ be the proportion of occurrences of word $j$ in contexts containing word $i$. Define $\Pr(i,j) \coloneqq \Pr(j|i)\Pr(i)$ to be the *skip-gram distribution*. We use the shorthand $P_{ij} \coloneqq \Pr(i,j)$ and $P_i \coloneqq \Pr(i)$.

Let $\boldsymbol{W} \in \mathbb{R}^{V \times d}$ be a trainable weight matrix whose $i^{\text{th}}$ row $\boldsymbol{w}_i$ is the $d$-dimensional embedding vector for word $i$. We restrict our focus to the underparameterized regime $d \ll V$, in accordance with practical settings. The goal is to imbue $\boldsymbol{W}$ with semantic structure so that the inner products between word embeddings capture semantic similarity. To do this, one often uses an iterative procedure that aligns frequently co-occurring words and repels unrelated words. The principle underlying this method is the *distributional hypothesis*, which posits that semantic structure in natural language can be discovered from the co-occurrence statistics of the words (Harris, 1954).

**Primer on `word2vec`.** In `word2vec` "skip-gram with negative sampling" (SGNS),[3] two embedding matrices $\boldsymbol{W}$ and $\boldsymbol{W}'$ are trained[4] using stochastic gradient descent to minimize the contrastive loss

$$\mathcal{L}_{\texttt{w2v}}(\boldsymbol{W}, \boldsymbol{W}') = \mathop{\mathbb{E}}_{i,j\sim\Pr(\cdot,\cdot)}\left[\Psi_{ij}^+ \log\left(1 + e^{-\boldsymbol{w}_i^\top \boldsymbol{w}_j'}\right)\right] + \mathop{\mathbb{E}}_{\substack{i\sim\Pr(\cdot)\\j\sim\Pr(\cdot)}}\left[\Psi_{ij}^- \log\left(1 + e^{\boldsymbol{w}_i^\top \boldsymbol{w}_j'}\right)\right]. \quad (1)$$

The averages are estimated by drawing samples from the corpus. The nonnegative hyperparameters $\{\Psi_{ij}^+\}$ and $\{\Psi_{ij}^-\}$ are reweighting coefficients for word pairs; we use them here to capture the effect of several of `word2vec`'s implementation details, including subsampling (i.e., probabilistically discarding frequent words during iteration), dynamic window sizes, and different negative sampling distributions. All of these can be seen as preprocessing techniques that directly modify the unigram and skip-gram distributions. In Appendix A.3 we provide more detail about these engineering tricks and discuss how one can encode their effects in $\Psi^+$ and $\Psi^-$.

The quality of the resulting embeddings are evaluated using standard semantic understanding benchmarks. For example, the Google analogy test set measures how well the model can complete analogies (e.g., man:woman::king:?) via vector addition (Mikolov et al., 2013). Importantly, this benchmark is distinct from the optimization task, and performing well on it requires representation learning.

The global minimizer of $\mathcal{L}_{\texttt{w2v}}$ is the *pointwise mutual information* (PMI) matrix

$$\operatorname*{arg\,min}_{\boldsymbol{w}_i^\top \boldsymbol{w}_j'} \mathcal{L}_{\texttt{w2v}}(\boldsymbol{W}, \boldsymbol{W}') = \log\left(\frac{\Psi_{ij}^+ P_{ij}}{\Psi_{ij}^- P_i P_j}\right), \quad (2)$$

where the minimization is over the inner products (Levy & Goldberg, 2014). Crucially, the PMI minimizer can only be realized ($\boldsymbol{W}\boldsymbol{W}'^\top = \mathrm{PMI}$) if there is no rank constraint ($d \geq \mathrm{rank}(\mathrm{PMI})$). This condition is always violated in practice. It is crucial, then, to determine *which* low-rank approximation of the PMI matrix is learned by `word2vec`. It is *not* the least-squares approximation; the resulting embeddings are known to perform significantly worse on downstream tasks such as analogy completion (Levy et al., 2015). This is because the divergence at $P_{ij}/P_i P_j \to 0$ causes least squares to over-allocate fitting power to these rarely co-occurring word pairs. Various alternatives have been proposed, including the *positive PMI*, $\mathrm{PPMI}_{ij} = \max(0, \mathrm{PMI}_{ij})$, but we find that these still differ from the embeddings learned by `word2vec`, both in character and in performance (Figure 3).

Our approach is different: rather than approximate the minimizer of $\mathcal{L}_{\texttt{w2v}}$, we obtain the *exact* minimizer of a (Taylor) approximation of $\mathcal{L}_{\texttt{w2v}}$. Though this may seem to be a coarser approximation, we are well-compensated by the ability to analytically treat the implicit bias of gradient descent, which enables us to give a full theory of *how* and *which* low-rank embeddings are learned.

---

[3]Throughout this paper, we use the abbreviated "word2vec" to refer to the `word2vec` SGNS algorithm.

[4]$\boldsymbol{W}$ is for "center" words, and $\boldsymbol{W}'$ is for "context" words. For simplicity, we consider the setting $\boldsymbol{W}' = \boldsymbol{W}$; in Figure 3 we show that this is sufficient for good performance on semantic understanding tasks.

# 3   Quadratic Word Embedding Models

We set $\boldsymbol{W}' = \boldsymbol{W}$ and study the quartic approximation of $\mathcal{L}_{\texttt{w2v}}$ around the origin:

$$\mathcal{L}(\boldsymbol{W}) \coloneqq \underset{i,j\sim\mathrm{Pr}(\cdot,\cdot)}{\mathbb{E}}\left[\Psi_{ij}^{+}\left(\frac{(\boldsymbol{w}_i^{\top}\boldsymbol{w}_j)^2}{4} - \boldsymbol{w}_i^{\top}\boldsymbol{w}_j\right)\right] + \underset{\substack{i\sim\mathrm{Pr}(\cdot)\\ j\sim\mathrm{Pr}(\cdot)}}{\mathbb{E}}\left[\Psi_{ij}^{-}\left(\frac{(\boldsymbol{w}_i^{\top}\boldsymbol{w}_j)^2}{4} + \boldsymbol{w}_i^{\top}\boldsymbol{w}_j\right)\right]. \quad (3)$$

Note that the key quantities are the inner products between embeddings. There is no privileged coordinate basis in embedding space. Since the objective is quadratic in these inner products, we refer to the resulting models *quadratic word embedding models* (QWEMs).[5]

**Model equivalence classes.**   Since $\mathcal{L}(\boldsymbol{W})$ is invariant under orthogonal transformations of the right singular vectors of $\boldsymbol{W}$, we define the right orthogonal equivalence class

$$\mathrm{REquiv}(\boldsymbol{W}) \coloneqq \left\{\boldsymbol{W}\boldsymbol{U} \mid \boldsymbol{U} \in \mathbb{R}^{d\times d}, \boldsymbol{U}^{\top}\boldsymbol{U} = \boldsymbol{I}\right\}. \quad (4)$$

## 3.1   QWEM is equivalent to matrix factorization with square loss.

**Target matrix.**   We start by introducing a matrix $\boldsymbol{M}^{\star}$. We will show in Theorem 1 that $\boldsymbol{M}^{\star}$ is the optimization target for QWEM, just as the PMI matrix is the optimization target for `word2vec`.

$$\boldsymbol{M}_{ij}^{\star} \coloneqq \frac{\Psi_{ij}^{+}P_{ij} - \Psi_{ij}^{-}P_iP_j}{\frac{1}{2}(\Psi_{ij}^{+}P_{ij} + \Psi_{ij}^{-}P_iP_j)}. \quad (5)$$

To understand this quantity, first note that if language were a stochastic process with independently sampled words, the co-occurrence statistics would be structureless, i.e., $P_{ij} - P_iP_j = 0$. The distributional hypothesis then suggests that algorithms may learn semantics by modeling the statistical deviations from independence. It is exactly these (relative) deviations that comprise the optimization target $\boldsymbol{M}^{\star}$ and serve as the central statistics of interest in our theory. Furthermore, $\boldsymbol{M}^{\star}$ can be seen as an approximation the PMI matrix; see Appendix D.

**Reweighting hyperparameters.**   Our goal now is to directly convert Equation (3) into a matrix factorization problem. To do this, we make some judicious choices for the hyperparameters. We first define the quantity $\boldsymbol{G}_{ij} \coloneqq \Psi_{ij}^{+}P_{ij} + \Psi_{ij}^{-}P_iP_j$, which captures the aggregate effect of the reweighting hyperparameters on the optimization. Then we establish the following hyperparameter setting.

**Setting 3.1** (Symmetric $\Psi^{+}, \Psi^{-}$ and constant $\boldsymbol{G}_{ij}$). Let $\Psi_{ij}^{+} = \Psi_{ji}^{+}$ and $\Psi_{ij}^{-} = \Psi_{ji}^{-}$ so that, by symmetry, the eigendecomposition $\boldsymbol{M}^{\star} = \boldsymbol{V}^{\star}\boldsymbol{\Lambda}\boldsymbol{V}^{\star\top}$ exists. Let $\boldsymbol{G}_{ij} = g$ for some constant $g$.

Note that infinitely many choices of $\Psi^{+}$ and $\Psi^{-}$ are encompassed by this setting. Let us study a concrete example: $\Psi_{ij}^{+} = \Psi_{ij}^{-} = (P_{ij} + P_iP_j)^{-1}$, so that $\boldsymbol{G}_{ij} = g = 1$. This has the effect of down-weighting frequently appearing words and word pairs, which hastens optimization and prevents the model from over-allocating fitting power to words such as "the" or "and" which may not individually carry much semantic content. This is exactly the justification given for subsampling in `word2vec`, which motivates Setting 3.1. Indeed, in Appendix A.3 we discuss how these choices of $\Psi^{+}$ and $\Psi^{-}$ can be seen as approximating several of the implementation details and engineering tricks in `word2vec`. In Figures 2 and 3, we show that this simplified hyperparameter setting does not wash out the relevant structure, thus retaining realism.

With these definitions, we state our key result: rank-constrained quadratic word embedding models trained under Setting 3.1 learn the top $d$ eigendirections of $\boldsymbol{M}^{\star}$.

**Theorem 1** (QWEM = unweighted matrix factorization). *Under Setting 3.1, the contrastive loss Equation (3) can be rewritten as the unweighted matrix factorization problem*

$$\mathcal{L}(\boldsymbol{W}) = \frac{g}{4}\left\|\boldsymbol{W}\boldsymbol{W}^{\top} - \boldsymbol{M}^{\star}\right\|_{\mathrm{F}}^{2} + \mathrm{const}. \quad (6)$$

*If $\boldsymbol{\Lambda}_{[:d,:d]}$ is positive semidefinite, then the set of global minima of $\mathcal{L}$ is given by*

$$\underset{\boldsymbol{W}}{\arg\min}\,\mathcal{L}(\boldsymbol{W}) = \mathrm{REquiv}\left(\boldsymbol{V}_{[:,:d]}^{\star}\boldsymbol{\Lambda}_{[:d,:d]}^{1/2}\right). \quad (7)$$

*Proof.* Equation (6) follows from completing the square in Equation (3). Equation (7) follows from the Eckart-Young-Mirsky theorem. In Appendix A.2, we note that the PSD assumption is easily satisfied in practice.

---

[5]We use "QWEM" as shorthand for minimizing the quartic approximation of the `word2vec` loss, and "QWEMs" for the resulting embeddings. We emphasize that QWEMs do not refer to a new model architecture.

We emphasize that although previous results prove that word embedding models find low-rank approximations to some target matrix (e.g., PMI for `word2vec`), previous results do not establish *which* low-rank factorization is learned. To our knowledge, our result is the first to solve for the rank-constrained minimizer of a practical word embedding task. Furthermore, our solution is given in terms of only the corpus statistics and the hyperparameters $\Psi^+$ and $\Psi^-$. In particular, we show that QWEMs learn compressed representations of the relative deviations between the true co-occurrence statistics and the baseline of independently distributed words. Unlike previous work, we do not require stringent assumptions on the data distribution (e.g., spherically symmetric latent vectors, etc.).

The main limitation of Theorem 1 is its restriction to Setting 3.1. What happens in the general case?

**Proposition 2.** *Equation (3) can be rewritten as the weighted matrix factorization problem*

$$\mathcal{L}(\boldsymbol{W}) = \frac{1}{4} \sum_{i,j} \boldsymbol{G}_{ij} (\boldsymbol{W}\boldsymbol{W}^\top - \boldsymbol{M}^\star)^2_{ij} + \text{const.} \tag{8}$$

*Let $\Psi^+$ and $\Psi^-$ each be symmetric in $i, j$, let $\boldsymbol{g} \in \mathbb{R}^V$ be an arbitrary vector with non-negative elements, and let $\boldsymbol{\Gamma} = \operatorname{diag}(\boldsymbol{g})^{1/2}$. Then the eigendecomposition of $\boldsymbol{\Gamma}\boldsymbol{M}^\star\boldsymbol{\Gamma} = \boldsymbol{V}_\Gamma^\star \boldsymbol{\Lambda}_\Gamma \boldsymbol{V}_\Gamma^{\star\top}$ exists. If $\boldsymbol{G} = \boldsymbol{g}\boldsymbol{g}^\top$ and $\boldsymbol{\Lambda}_{[:d,:d]}$ is positive semidefinite, then the set of global minima of $\mathcal{L}$ is given by*

$$\arg\min_{\boldsymbol{W}} \mathcal{L}(\boldsymbol{W}) = \text{REquiv}\left(\boldsymbol{\Gamma}^{-1} \boldsymbol{V}_{\Gamma,[:,:d]}^\star \boldsymbol{\Lambda}_{\Gamma,[:d,:d]}^{1/2}\right). \tag{9}$$

*Proof.* See Appendix C. Equation (8) says that in the general case, the form of the target matrix remains unchanged. Equation (9) says that if the hyperparameters are chosen so that $\boldsymbol{G}$ is rank-1, then we can characterize the minimizer. However, if $\boldsymbol{G}$ is not rank-1, then we do not know what low-rank factorization is learned, nor can we describe the training dynamics; indeed, weighted matrix factorization is known to be NP-hard (Gillis & Glineur, 2011). For this reason, we do not revisit these more general settings, and we hereafter restrict our focus to Setting 3.1. This is ultimately justified by the empirically close match between our theory and the embeddings learned by `word2vec` (Figure 3).

### 3.2 Training dynamics of QWEMs reveal implicit bias towards low rank.

Note that despite the simplification afforded by Theorem 1, the minimization problem Equation (6) is still nonconvex since $\mathcal{L}(\boldsymbol{W})$ is quartic in $\boldsymbol{W}$. Thus, there remain questions regarding convergence time and the effect of early stopping regularization. To study them, we examine the training trajectories induced by gradient flow. The central variables of our theory will be the economy-sized SVD of the embeddings, $\boldsymbol{W}(t)^\top = \boldsymbol{U}(t)\boldsymbol{S}(t)\boldsymbol{V}(t)^\top$, and the eigendecomposition of the target, $\boldsymbol{M}^\star = \boldsymbol{V}^\star\boldsymbol{\Lambda}\boldsymbol{V}^{\star\top}$. For convenience, we define $\lambda_k := \boldsymbol{\Lambda}_{kk}$ and $s_k := \boldsymbol{S}_{kk}$.

In general, it is difficult to solve the gradient descent dynamics of Equation (3) from arbitrary initialization. We first consider a simple toy case in which the initial embeddings are already aligned with the top $d$ eigenvectors of $\boldsymbol{M}^\star$.

**Lemma 3.1** (Training dynamics, aligned initialization)**.** *If $\tilde{\boldsymbol{\Lambda}} := \boldsymbol{\Lambda}_{[:d,:d]}$ is positive semidefinite and $\boldsymbol{V}(0) = \boldsymbol{V}_{[:,:d]}^\star$, then under Setting 3.1, the gradient flow dynamics $\frac{\mathrm{d}\boldsymbol{W}}{\mathrm{d}t} = -\frac{1}{2g}\nabla\mathcal{L}(\boldsymbol{W})$ yields*

$$\boldsymbol{U}(t) = \boldsymbol{U}(0) \tag{10}$$

$$\boldsymbol{S}(t) = \boldsymbol{S}(0)\left(e^{-\tilde{\boldsymbol{\Lambda}}t} + \boldsymbol{S}^2(0)\tilde{\boldsymbol{\Lambda}}^{-1}\left(\boldsymbol{I} - e^{-\tilde{\boldsymbol{\Lambda}}t}\right)\right)^{-1/2} \tag{11}$$

$$\boldsymbol{V}(t) = \boldsymbol{V}(0) = \boldsymbol{V}_{[:,:d]}^\star \tag{12}$$

The proof is given in Appendix C. See Saxe et al. (2014) for a proof in an equivalent learning problem.

This result states that the final embeddings' PCA directions are given by the top $d$ eigenvectors of $\boldsymbol{M}^\star$, that the dynamics are decoupled in this basis, and that the population variance of the embeddings along the $k^{\text{th}}$ principal direction increases sigmoidally from $s_k^2(0)$ to $\lambda_k$ in a characteristic time $t = \tau_k := (1/\lambda_k)\ln\left(\lambda_k/s_k^2(0)\right)$. These training dynamics have been discovered in a variety of other learning problems (Saxe et al., 2014; Gidel et al., 2019; Atanasov et al., 2022; Simon et al., 2023b; Dominé et al., 2023). Our result adds self-supervised quadratic word embedding models to the list.

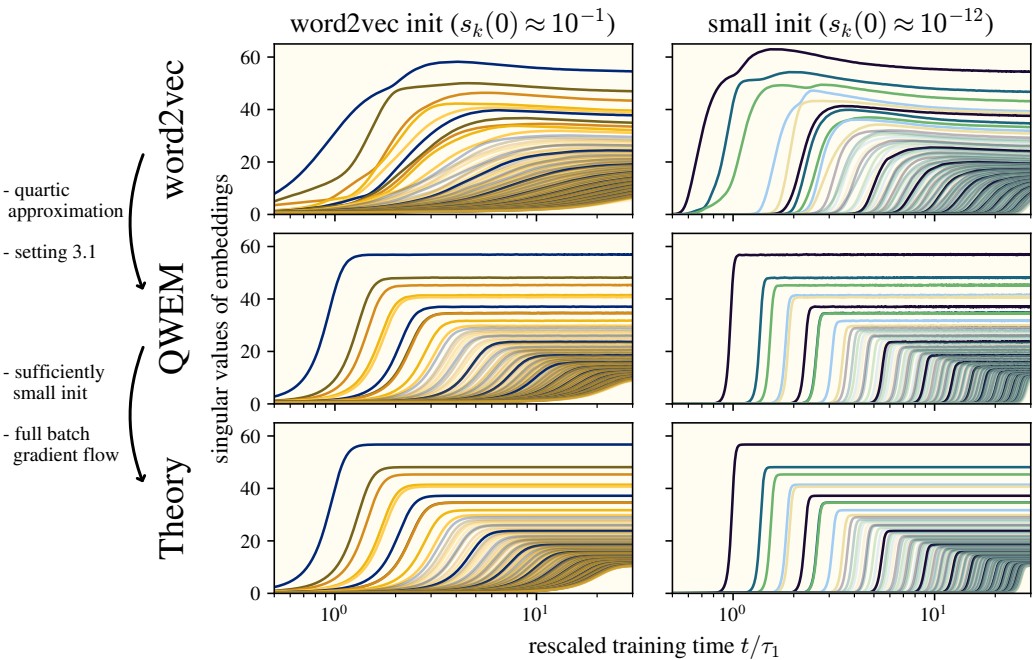

**Figure 2: Theory matches experiment.** We make two simplifications to the `word2vec` algorithm: a quartic approximation of the loss, and a restriction on the reweighting hyperparameters. We train these QWEMs on 2 billion tokens of English Wikipedia (see Appendix A for details) and compare to `word2vec`. We find good qualitative match in the singular value dynamics, both with the standard `word2vec` initialization scheme and with small random initialization. (For evidence that the singular vectors match as well, see Figure 3.) We compare the dynamics to the prediction of Result 3, which is derived in the vanishing initialization limit with full-batch gradient flow. Even though the experiment uses stochastic mini-batching, non-vanishing learning rate, and large initialization, we find excellent agreement even up to constant factors.

It is restrictive to require perfect alignment $V = V^\star$ at initialization. To lift this assumption, we show that in the limit of vanishing random initialization, the dynamics are indistinguishable from the aligned case, up to orthogonal transformations of the right singular vectors of $W$. To our knowledge, previous works have not derived this equivalence for under-parameterized matrix factorization.

**Result 3** (Training dynamics, vanishing random initialization)**.** *Initialize $W_{ij}(0) \sim \mathcal{N}(0, \sigma^2)$, and let $S(0)$ denote the singular values of $W(0)$. Define the characteristic time $\tau_1 := \lambda_1^{-1} \ln(\lambda_1/\sigma^2)$ and the rescaled time variable $\tilde{t} = t/\tau_1$. Define $W^\star(0) := V^\star_{[:,:d]} S(0)$ and let $W^\star(t)$ be its gradient flow trajectory given by Lemma 3.1. If $\Lambda_{[:d,:d]}$ is positive semidefinite, then under Setting 3.1 we have with high probability that*

$$\forall \tilde{t} \geq 0, \quad \lim_{\sigma^2 \to 0} \min_{W' \in \mathrm{REquiv}(W(\tilde{t}\tau_1))} \|W' - W^\star(\tilde{t}\tau_1)\|_{\mathrm{F}} = 0. \tag{13}$$

*Derivation.* See Appendix C. The main idea is to study the dynamics of the $QR$ factorization of $W^\top V^\star$. We write the equation of motion, discard terms that become small in the limit, solve the resulting equation, and show that the discarded terms remain small. We conjecture that our argument can be made rigorous by appropriately bounding the discarded terms. We leave this to future work.

Result 3 generalizes previous work that establishes a *silent alignment* phenomenon for linear networks with scalar outputs (Atanasov et al., 2022). Here, for all $k \leq d$, $V_{[:,:k]}$ quickly aligns with $V^\star_{[:,:k]}$ while $s_k$ remains near initialization; thus the alignment assumption is quickly near-satisfied and Lemma 3.1 becomes applicable. We conclude that quadratic word embedding models trained from small random initialization are inherently greedy spectral methods. The principal components of the embeddings learn a one-to-one correspondence with the eigenvectors of $M^\star$, and each component is realized independently and sequentially. Thus, early stopping acts as an implicit regularizer, constraining model capacity in terms of rank rather than weight norm.

| word2vec feature neighbors | QWEM feature neighbors | svd(PPMI) feature neighbors |
|---|---|---|
| **(PC1)** jones scott gary frank robinson terry michael david eric kelly | **(PC1)** tom jones david frank michael scott robinson kelly tony | **(PC1)** eric cooper jones dennis oliver sam tom robinson |
| **(PC2)** government council national established in state united republic | **(PC2)** government council establishment appointed republic union | **(PC2)** jones dennis eric robinson scott michael oliver taylor |
| **(PC3)** adjacent located built surface powered bay meters road near | **(PC3)** bay adjacent located north junction southwest road northeast | **(PC3)** government establishment foreign authorities leaders |
| **(PC11)** combat enemy offensive deployed naval artillery narrative | **(PC10)** offensive combat war artillery enemy naval defensive | **(PC11)** deployed force combat patrol command naval squadron |
| **(PC12)** whilst trained skills competitions studying teaching artistic | **(PC11)** trained skills whilst studying competitions aged honours | **(PC14)** piano vocal orchestra solo music instrumental recordings |
| **(PC13)** britain produced anglo ltd welsh australian scottish sold | **(PC12)** britain produced considerable price notably industry sold | **(PC15)** england thus great price meant liverpool share earl enjoyed |
| **(PC100)** il northern di worker laid contributions ireland down oak | **(PC100)** doctor bar lives disc oregon credited ultimate split serial | **(PC100)** org figure standing riding with http green date www parent |

**(a)** Words with smallest cosine distance to embedding principal components

|  | word2vec | QWEM | svd($M^\star$) | svd(PPMI) | svd(PMI) |
|---|---|---|---|---|---|
| Google Analogies | 68.0% | 65.1% | 66.3% | 50.6% | 8.4% |
| MEN test set | 0.744 | 0.755 | 0.755 | 0.740 | 0.448 |
| WordSim353 | 0.698 | 0.682 | 0.683 | 0.690 | 0.221 |

**(b)** Performance on standard word embedding benchmarks

**Figure 3: QWEMs and word2vec learn similar features, whereas PMI (and variants) differ qualitatively.**
For the following, all models $W \in \mathbb{R}^{V \times d}$ are trained with $V = 10,000$ and $d = 200$ on 2 billion tokens of Wikipedia. See Appendix A for experimental details. We denote $\mathrm{svd}(M) \coloneqq \arg\min_W \|WW^\top - M\|_\mathrm{F}^2$. **(Top.)** We compute the principal components of the final embeddings and list the closest embeddings. See Appendix B for a quantitative plot of subspace overlaps. Top section: top three components of word2vec and QWEM represent topic-level concepts (biography, government, geography) corresponding to common topics on Wikipedia. Middle section: components closest to word2vec components 11, 12, 13. Up to reordering, QWEMs match closely and remain interpretable, while positive PMI deviates. Bottom row: late components lose their interpretability. **(Bottom.)** Since QWEMs and word2vec learn similar features, they perform similarly on the vector addition analogy completion task. Explicit factorization of $M^\star$ almost matches the performance of word2vec, much better than the best previously-known SVD embeddings.

## 3.3 Empirically, QWEMs are a good proxy for word2vec.

Our rationale for studying QWEMs is to gain analytical tractability (e.g., Result 3) in a setting that is "close to" the true setting of interest: word2vec. In Figure 3, we check whether QWEMs are in fact a faithful representative for word2vec. We find that not only do QWEMs nearly match word2vec on the analogy completion task, they also learn very similar representations, as measured by the alignment between their singular vectors. Importantly, QWEMs are closer to word2vec than the least-squares approximations of either PMI or PPMI. This underscores the importance of accounting for the implicit bias of gradient descent.

To run the experiments in Figures 1 to 3, we implemented a GPU-enabled training algorithm for both word2vec and QWEM. The corpus data is streamed from the hard drive in chunks to avoid memory overhead and excessive disk I/O, and the loops for batching positive and negative word pairs are compiled for efficiency. Notably, it is often faster to construct and explicitly factorize $M^\star$ (e.g., with vocabulary size $V = 10,000$, it takes $\sim 10$ minutes in total on a single GeForce GTX 1660 GPU).

Figure 3 suggests that if one is interested in understanding the learning behaviors of word2vec, it suffices to study QWEMs. Then Theorem 1 and Result 3 state that if one is interested in understanding the learning behaviors of QWEMs, it suffices to study $M^\star$. In this spirit, we will now study $M^\star$ to investigate certain aspects of representation learning in word embedding models.

# 4 Linear Structure in Latent Space

Result 3 reveals that, for QWEMs, the "natural" basis of the learning dynamics is simply the eigenbasis of $\boldsymbol{M}^\star$. Figure 3 suggests that this basis already encodes concepts interpretable to humans. These are the fundamental features learned by the model: each word embedding is naturally expressed as a linear combination of these orthogonal latents. Both early stopping and small embedding dimension regularize the model by constraining the number (but not the character) of available latents.

We may conclude that natural language contains linear semantic structure in its co-occurrence statistics, and that it is easily extracted by word embedding algorithms. It is not unreasonable to expect, then, that other semantic concepts may be encoded as linear subspaces. This idea is the *linear representation hypothesis*, and it motivates modern research in more sophisticated language models, including representation learning (Park et al., 2023; Jiang et al., 2024; Wang et al., 2024), mechanistic interpretability (Li et al., 2023; Nanda et al., 2023; Lee et al., 2024), and LLM alignment (Lauscher et al., 2020; Zou et al., 2023; Li et al., 2024). To make these efforts more precise, it is important to develop a quantitative understanding of these linear representations in simple models. Our Theorem 1 provides a new lens for this analysis: we may gain insight by studying the properties of $\boldsymbol{M}^\star$.

In word embedding models, a natural category of abstract linear representations are the difference vectors between semantic pairs, e.g., $\{\boldsymbol{r}^{(n)}\}_n = \{\textbf{man} - \textbf{woman}, \textbf{king} - \textbf{queen}, \dots\}$ for gender binaries. If the $\boldsymbol{r}^{(n)}$ are all approximately equal, then the model can effectively complete analogies via vector addition (see Appendix A.4). Following Ilharco et al. (2022), we call the $\boldsymbol{r}^{(n)}$ *task vectors*.

Here, we provide empirical evidence for the following dynamical picture of learning. The model internally builds task vectors in a sequence of noisy learning steps, and the geometry of the task vectors is well-described by a spiked random matrix model. Early in training, semantic signal dominates; however, later in training, noise may begin to dominate, causing a degradation of the model's ability to resolve the task vector. We validate this picture by studying the task vectors in a standard analogy completion benchmark.

We emphasize that, due to our Result 3, any result comparing the final embeddings of many models of different sizes $d$ is *fully equivalent* to a result considering the time course of learning for a single (sufficiently large) model. Therefore, although we vary the model dimension in our plots, these can be viewed as results concerning the dynamics of learning in word embedding models.

**Task vectors are often concentrated on a few dominant eigen-features.** Task vectors derived from the analogy dataset are neither strongly aligned with a single model eigen-feature, nor are they random vectors. Instead, they exhibit *localization*: a handful of the top eigen-features are primarily responsible for the task vector. In some cases, these dominant eigen-features are interpretable and clearly correlate with the abstract semantic concept associated with the task vector (Figure 4).

**Task vectors are well-described by a spiked random matrix model.** To study the geometry of the task vectors within a class of semantic binaries, we consider stacking task vectors to produce the matrix $\boldsymbol{R}_d \in \mathbb{R}^{N \times d}$, where $N$ is the number of word pairs and $d$ is the embedding dimension. We note that $\boldsymbol{R}_d$ can be computed in closed form using our Theorem 1. If all the task vectors align (as desired for analogy completion), then $\boldsymbol{R}_d$ will be a rank-1 matrix; if the task vectors are all unrelated, then $\boldsymbol{R}_d$ will have a broad spectrum. This observation suggests that a useful object to study is the empirical spectrum of the Gram matrix $\boldsymbol{G}_d := \boldsymbol{R}_d \boldsymbol{R}_d^\top \in \mathbb{R}^{N \times N}$.

We find that this spectrum is very well-described by the *spiked covariance model*, a well-known distribution of random matrices. In the model, one studies the spectrum of $\boldsymbol{Z} = \boldsymbol{\Xi}\boldsymbol{\Xi}^\top + \mu \boldsymbol{a}\boldsymbol{a}^\top$, where $\boldsymbol{\Xi}$ is a random matrix with i.i.d. mean-zero entries with variance $\sigma^2$ and $\mu \boldsymbol{a}\boldsymbol{a}^\top$ is a deterministic rank-1 perturbation with strength $\mu$. If $\mu$ is sufficiently large compared to $\sigma$, then in the asymptotic regime the spectrum of $\boldsymbol{Z}$ is known to approach the Marchenko-Pastur distribution with a single outlier eigenvalue (Baik et al., 2005). We consistently observe this structure in the real empirical data, across both model dimension and semantic families (Figure 4). We note that $N \approx 30$ is fairly small, and so it is somewhat surprising that results from the asymptotic regime visually appear to hold.

We interpret this observation as evidence that task vectors are well-described as being random vectors with a strong deterministic signal (e.g., Gaussian random vectors with nonzero mean). One expects that in a high-quality linear representation, the signal is simply the mean task vector, and that it overwhelms the random components. To understand whether a model can effectively utilize the task vectors, then, we examine the relative strength of the spike compared to the noisy bulk.

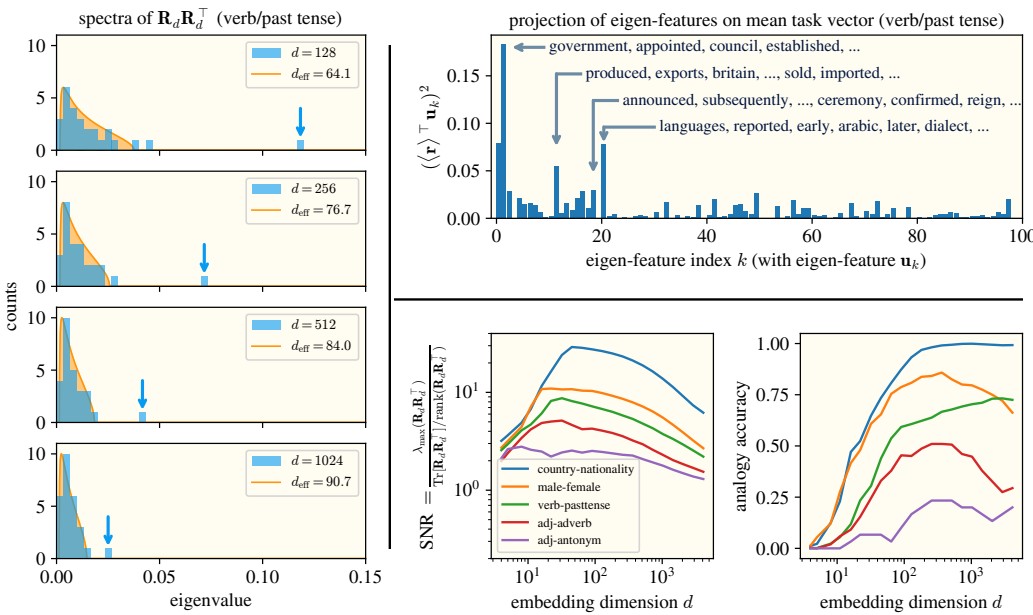

**Figure 4: Models build linear representations from a few informative and many noisy eigen-features.** In the left and upper plots, we examine task vectors between verb past tenses and their participle (e.g., **went** − **going**). In Appendix B, we show that these observations hold for other semantic binaries. **(Left.)** The spectrum of the Gram matrix (histogram) is well-described by a Marchenko-Pastur distribution (orange) plus an outlier "spike," across model sizes $d$. See Appendix A.6 for details. **(Top.)** The spike corresponds to the average task vector, which comprises a few dominant eigen-features. Many of these features correspond to concepts related to history or temporal change, consistent with this semantic category. **(Bottom.)** We measure the strength of the spike across model size $d$ for various semantic categories. We find that the spike strength correlates strongly with the model's ability to use the task vectors for analogy completion.

**The strength of the spike perturbation corresponds to the robustness of the task vector.** We measure the relative strength of the signal-containing spike using an empirical measure of the signal-to-noise ratio: $\mathrm{snr} := \lambda_{\max}(\boldsymbol{G}) \cdot \mathrm{rank}(\boldsymbol{G}) / \mathrm{Tr}[\boldsymbol{G}]$. This quantity is simply the ratio between the maximum eigenvalue (i.e., the signal strength) and the mean nonzero eigenvalue (i.e., the typical variation due to noise). We find that as the model learns representations of increasing rank during learning, it first captures an increasing fraction of the signal (consistent with the previous observation regarding the localization of the task vector). Later in training, the noise begins to dominate, and the model's ability to resolve the signal degrades. Finally, the maximum achieved SNR serves as a coarse predictor for how effectively the model can use the representation for downstream tasks: the model achieves higher analogy completion scores on semantic directions with higher SNR.

Together, these observations provide evidence that useful linear representations arise primarily from the model's ability to resolve signal-containing eigen-features without capturing excess noise from extraneous eigen-features. Furthermore, our results suggest that tools from random matrix theory may be fruitfully applied to understand and characterize this interplay. We leave this to future work.

## 5 FAQ

**Can the assumptions be relaxed?** Our theoretical results require only four assumptions: quartic approximation of the loss, our Setting 3.1, small initial weights, and population gradient flow. The latter two are technical conveniences that simplify the analysis; Figure 2 suggests that they may be relaxed (or possibly eliminated) with additional effort. The first two are genuine approximations of the word2vec algorithm; their validity is supported by our empirical checks (see Figures 1, 3 and 5 and Appendix A.3). To further understand why these two approximations are technically useful, and what happens if they are relaxed, we perform empirical ablation tests in Figure 7.[6]

---

[6] We note that these auxiliary experiments use a different training corpus, model size, and hyperparameter choices, indicating that our theory is not sensitive to these choices.

The clear stepwise learning and decoupled dynamics in our theory originate from Setting 3.1. This condition turns the weighted factorization problem into an unweighted factorization. As a consequence, the singular value/vector dynamics quickly become "untangled" in the early stage of training; see Proposition 2 and the subsequent discussion. Somewhat surprisingly, this simplification is "close enough" to word2vec. Though the word2vec singular vectors do exhibit mild "mixing" in time (Figure 2), the overall learning dynamics are well-described by the simplified setting. This agreement is partly because the recommended word2vec hyperparameters approximate satisfy Setting 3.1 (see Appendix A.3). Our results suggest that future efforts to understand more complex learning systems may find purchase in identifying useful approximations of this kind.

**Why do QWEM factorizations describe word2vec better than PMI factorizations?**  The crux of the issue lies in adequately handling the rank constraint. The idea behind factorizing PMI via SVD is to first solve the unconstrained minimizer of the loss (i.e., the PMI matrix) and then, at the end, apply the rank constraint by choosing the closest (in Frobenius norm) rank-$d$ matrix. This does not work well because the loss basin is extremely wide and shallow, so the global unconstrained minimum is actually very far from where the model actually ends up via gradient descent in finite time. We take a different approach: by first approximating the loss landscape itself, we can account for the rank constraint throughout the entire optimization trajectory. As a result, our prediction for what word2vec learns is significantly more accurate.

**Why does decreasing SNR sometimes still yield high performance in Figure 4?**  This is an ancillary effect of using top-1 accuracy as the performance metric. As a concrete example, consider the analogy "France : Paris :: Japan : Tokyo," satisfying japan + (paris − france) ≈ tokyo. As the effective embedding dimension increases, the SNR may decrease; at the same time, the typical separation between embeddings increases (due to the increased available volume in latent space). This means that there are fewer nearby "competitors" for tokyo. In the large $d$ regime, we find that the embeddings are sufficiently spaced, and tokyo is still the nearest embedding to japan + (paris − france) despite an increase in absolute error (see Figure 8). Thus, top-1 performance does not degrade with decreasing SNR. We note that unintuitive effects associated with using top-$k$ accuracy have been observed in LLMs as well (Schaeffer et al., 2024).

**How might one apply these results to other self-supervised learning tasks?**  Since our results are distribution agnostic, Theorem 1 can be extended to establish an *explicit equivalence* between self-supervised contrastive learning and supervised matrix factorization. In particular, if the contrastive objective has the functional form $\mathcal{L}(\boldsymbol{W}) = \mathbb{E}_{\text{positive pairs}}[f^+(\boldsymbol{w}^T\boldsymbol{w}')] + \mathbb{E}_{\text{negative pairs}}[f^-(\boldsymbol{w}^T\boldsymbol{w}')]$ for some differentiable $f^+(\cdot)$ and $f^-(\cdot)$, then one can simply take the quadratic Taylor approximation, complete the square, and obtain a target matrix in terms of the positive and negative distributions of the learning problem. See Appendix D.2 for a demonstration of this idea for the SimCLR loss. To obtain closed-form learning dynamics, one needs to find an equivalent of our Setting 3.1. This may be challenging depending on the form of the input data; for instance, when the inputs are not one-hot, it may require a whitened data covariance.

**What do these results tell us about feature learning?**  Two operational notions of *feature learning* are 1) optimization trajectories must escape the local vicinity of the (typically small) initialization (Yang & Hu, 2021; Jacot et al., 2021; Zhu et al., 2022; Atanasov et al., 2024; Kunin et al., 2025), and 2) learned weight matrices must project data onto target-relevant subspaces (Damian et al., 2022; Radhakrishnan et al., 2024). Our message complements this line of research by offering a practical yet solvable setting in which both behaviors appear: word2vec escapes its near-isotropic initialization region to learn a dense compression of the relative excess co-occurrence probability $\boldsymbol{M}^\star$, thereby aligning the embedding geometry with the most salient semantic linear subspaces. Our result is thus a step forward in the broader scientific project of obtaining quantitative, predictive descriptions of learning in practical algorithms.

**Limitations.**  Our results are limited to the symmetric setup (tied encoder/decoder weights, i.e., $\boldsymbol{W} = \boldsymbol{W}'$). We did not train and evaluate at scales that are considered state-of-the-art, nor did we compare against other embedding models such as GloVe (Pennington et al., 2014).

**Author contributions.** DK developed the analytical results, ran all experiments, and wrote the manuscript with input from all authors. JS proposed the initial line of investigation and provided insight at key points in the analysis. YB and MD helped shape research objectives and gave oversight and feedback throughout the project's execution.

**Acknowledgements.** DK is grateful to Daniel Kunin and Carl Allen for early conversations about learning linear analogy structure from small initialization. DK thanks Google DeepMind and BAIR for funding support and compute access. MRD thanks the U.S. Army Research Laboratory and the U.S. Army Research Office for supporting this work under Contract No. W911NF-20-1-0151.

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

# A  Experimental details

All our implementations use `jax` (Bradbury et al., 2018). In all reported experiments we use a vocabulary size $V = 10^4$, model dimension $d = 200$, and a context length $L = 32$ neighbors for each word. These were chosen fairly arbitrarily and our robustness checks indicated that our main empirical results were not sensitive to these choices.

## A.1  Training corpus

We train all word embedding models on the November 2023 dump of English Wikipedia, downloaded from `https://huggingface.co/datasets/wikimedia/wikipedia`. We tokenize by replacing all non-alphabetical characters (including numerals and punctuation) with whitespace and splitting by whitespace. The full corpus contains 6.4 million articles and 4.34 billion tokens in total. The number of tokens per article follows a long-tailed distribution: the (50%, 90%, 95%, 99%) quantiles of the article token counts are (364, 1470, 2233, 5362) tokens. We discard all articles with fewer than 500 tokens, leaving a training corpus of 1.58 million articles with 2.00 billion tokens in total.

We use a vocabulary consisting of the $V = 10^4$ most frequently appearing words. We discard out-of-vocabulary words from both the corpus and the benchmarks. Our robustness checks indicated that as long as the corpus is sufficiently large (as is the case here), it does not matter practically whether out-of-vocabulary words are removed or simply masked. This choice of $V$ retains 87% of the tokens in the training corpus and 53% of the analogy pairs.

## A.2  Optimization

We optimize using stochastic gradient descent with no momentum nor weight decay. Each minibatch consists of 100,000 word pairs (50,000 positive pairs and 50,000 negative pairs). For benchmark evaluations (Figure 3) we use a stepwise learning rate schedule in which the base learning rate is decreased by 90% at $t = 0.75t_{\max}$. We found that this was very beneficial, especially for QWEMs, which appear more sensitive to large learning rates. The finite-stepsize gradient descent dynamics of matrix factorization problems remains an interesting area for future research.

We directly train the tensor of embedding weights $\boldsymbol{W} \in \mathbb{R}^{d \times V}$. One potential concern is that since the model $\boldsymbol{W}\boldsymbol{W}^\top$ is positive semidefinite, it cannot reconstruct the eigenmodes of $\boldsymbol{M}^\star \in \mathbb{R}^{V \times V}$ with negative eigenvalue. However, we empirically find that with $V = 10^4$, $\boldsymbol{M}^\star$ has 4795 non-negative eigenvalues. Therefore, in the underparameterized regime $d \ll V$, the model lacks the capacity to attempt fitting the negative eigenmodes. Thus, the PSD-ness of our model poses no problem.

## A.3  Reweighting hyperparameters

Taking from the original `word2vec` implementation, we use the following engineering tricks to improve performance.

**Dynamic context windows.**  Rather than using a fixed context window length $L$, `word2vec` uniformly samples the context length between 1 and $L$ at each center word. In aggregate, this has the effect of more frequently sampling word pairs with less separation. Let $d_{ij}$ be the mean distance between words $i$ and $j$, when found co-occurring in contexts of length $L$. (Thus, $d_i$ is small for the words "phase" and "transition" since they are a linguistic collocation, but large for "proved" and "derived" since verbs are typically separated by many words.) Then it is not hard to calculate the effect of uniformly-distributed dynamic context window lengths; it is equivalent to setting

$$\Psi_{ij}^+ = \frac{L - d_{ij}}{\sum_{i'j'}(L - d_{i'j'})P_{i'j'}}. \tag{14}$$

**Subsampling frequent words.**  Mikolov et al. (2013) suggest discarding very frequent words with a frequency-dependent discard probability. This is akin to rejection sampling, with the desired unigram distribution flatter than the true Zipfian distribution. The original `word2vec` implementation uses an acceptance probability of

$$P_{acc}(i) = \min\left(1, \frac{10^{-3}}{P_i} + \sqrt{\frac{10^{-3}}{P_i}}\right). \tag{15}$$

This is equivalent to setting the hyperparameters

$$\Psi_{ij}^+ = \Psi_{ij}^- = P_{acc}(i)P_{acc}(j). \tag{16}$$

**Different negative sampling rate.** Finally, Mikolov et al. (2013) draw negative samples from a different distribution (empirically, they find that $P_j^{3/4}$ is particularly performant) and upweight the negative sampling term by a hyperparameter $k$ (they recommend $k \approx 5$ for large corpora). This is equivalent to setting

$$\Psi_{ij}^- = \frac{kP_j^{-1/4}}{V^{-1}\sum_{j'} P_{j'}^{-1/4}}. \tag{17}$$

Note that this is an asymmetric choice, so it is not covered by Setting 3.1. However, both this and the previous subsampling technique are methods for modifying and flattening the word distributions, as seen by the training algorithm. Our Setting 3.1 accomplishes the same thing. We speculate that these manipulations accomplish for language data what spectral whitening does for image data.

Taken together, these give a prescription for cleanly and concisely capturing several of the implementation details of `word2vec` in a set of training hyperparameters. In our experiments, when training `word2vec`, we use all of these settings – they can be combined by multiplying the different $\Psi^+$ and likewise for $\Psi^-$. For QWEMs, we use $\Psi_{ij}^+ = \Psi_{ij}^- = (P_{ij} + P_i P_j)^{-1}$, and modify $\Psi^+$ by combining with the setting for dynamic context windows. For fair comparison, we use $k = 1$ for the SGNS experiments in the main text.

### A.4 Benchmarks

We use the Google analogies described in Mikolov et al. (2013) for the analogy completion benchmark, `https://github.com/tmikolov/word2vec/blob/master/questions-words.txt`. We then compute the analogy accuracy using

$$\mathrm{acc}(\boldsymbol{W}) \coloneqq \frac{1}{|\mathcal{D}|} \sum_{(\boldsymbol{a},\boldsymbol{b},\boldsymbol{a}',\boldsymbol{b}')\in\mathcal{D}} \mathbf{1}_{\{\boldsymbol{b}'\}} \left( \underset{\boldsymbol{w}\in\{\boldsymbol{W}\}\setminus\{\boldsymbol{a},\boldsymbol{b},\boldsymbol{a}'\}}{\arg\max} \hat{\boldsymbol{w}}^\top \left( \hat{\boldsymbol{a}}' + (\hat{\boldsymbol{b}} - \hat{\boldsymbol{a}}) \right) \right) \tag{18}$$

$$= \frac{1}{|\mathcal{D}|} \sum_{(\boldsymbol{a},\boldsymbol{b},\boldsymbol{a}',\boldsymbol{b}')\in\mathcal{D}} \mathbf{1}_{\{\boldsymbol{b}'\}} \left( \underset{\boldsymbol{w}\in\{\boldsymbol{W}\}\setminus\{\boldsymbol{a},\boldsymbol{b},\boldsymbol{a}'\}}{\arg\min} \left\| \hat{\boldsymbol{a}} - \hat{\boldsymbol{b}} - \hat{\boldsymbol{a}}' + \hat{\boldsymbol{w}} \right\|_{\mathrm{F}}^2 \right), \tag{19}$$

where the 4-tuple of embeddings $(\boldsymbol{a}, \boldsymbol{b}, \boldsymbol{a}', \boldsymbol{b}')$ constitute an analogy from the benchmark data $\mathcal{D}$, hats (e.g., $\hat{\boldsymbol{w}}$) denote normalized unit vectors, $\mathbf{1}$ is the indicator function, and $\{\boldsymbol{W}\}$ is the set containing the word embeddings. The first expression measures cosine alignment between embeddings and the "expected" representation obtained by summing the query word and the task vector. The second expression measures the degree to which four embeddings form a closed parallelogram (Rumelhart & Abrahamson, 1973). The two forms of the accuracy are equivalent since

$$\underset{\boldsymbol{w}}{\arg\min} \left\| \hat{\boldsymbol{a}} - \hat{\boldsymbol{b}} - \hat{\boldsymbol{a}}' + \hat{\boldsymbol{w}} \right\|_{\mathrm{F}}^2 = \underset{\boldsymbol{w}}{\arg\min} \left( 2\hat{\boldsymbol{w}}^\top(\hat{\boldsymbol{a}} - \hat{\boldsymbol{b}} - \hat{\boldsymbol{a}}') + \mathrm{const.} \right) \tag{20}$$

$$= \underset{\boldsymbol{w}}{\arg\max} \hat{\boldsymbol{w}}^\top(\hat{\boldsymbol{a}}' + \hat{\boldsymbol{b}} - \hat{\boldsymbol{a}}). \tag{21}$$

To understand the role of normalization, we compared this metric with one in which only the candidate embedding is normalized:

$$\mathrm{winner} = \underset{\boldsymbol{w}\in\{\boldsymbol{W}\}\setminus\{\boldsymbol{a},\boldsymbol{b},\boldsymbol{a}'\}}{\arg\max} \hat{\boldsymbol{w}}^\top \left( \boldsymbol{a}' + (\boldsymbol{b} - \boldsymbol{a}) \right). \tag{22}$$

We found that the accuracy metric using this selection criterion yields performance measurements that are nearly identical to those given by the standard fully-normalized accuracy metric. We conclude that the primary role of embedding normalization is to prevent longer $\boldsymbol{w}$'s from "winning" the $\arg\max$ just by virtue of their length. The lengths of $\boldsymbol{a}, \boldsymbol{b}, \boldsymbol{a}'$ are only important if there is significant *angular* discrepancy between $(\hat{\boldsymbol{a}}' + \hat{\boldsymbol{b}} - \hat{\boldsymbol{a}})$ and $(\boldsymbol{a}' + \boldsymbol{b} - \boldsymbol{a})$; in the high-dimensional regime with relatively small variations in embedding length, we expect such discrepancies to be negligible.

For semantic similarity benchmarks, we use the MEN dataset (Bruni et al., 2014) and the WordSim353 dataset (Finkelstein et al., 2001). These datasets consist of a set of $N$ word pairs ranked by humans in order of increasing perceived semantic similarity. The model rankings are generated by computing the inner product between the embeddings in each pair and sorting them. The model is scored by computing the Spearman's rank correlation coefficient between its rankings and the human rankings:

$$\rho(\boldsymbol{W}) \coloneqq \mathrm{Corr}[r_{\mathrm{human}}(i), r_{\mathrm{model}}(i)] \tag{23}$$

where $r(i)$ is the rank of pair $i$ and $\mathrm{Corr}$ is the standard Pearson's correlation coefficient.

### A.5 Computational resources

Our implementations are relatively lightweight. The models can be trained on a home desktop computer with an i7 4-core processor, 32GB RAM, and a consumer-grade NVIDIA GTX 1660 graphics card, in about 2 hours. For the experiments in Figure 3, we train for 12 hours with a much lower learning rate. Our code is publicly available at `https://github.com/dkarkada/qwem`.

### A.6 Empirics of task vectors

We construct the theoretical embeddings $W$ by constructing $M^\star$ according to Equation (5), diagonalizing it, and applying Theorem 1. Due to right orthogonal symmetry, we are free to apply any right orthogonal transformation; we choose the identity as the right singular matrix, so that the components of each embedding in the canonical basis are simply its projections on the eigen-features. With this setup, it is easy to extract the embeddings resulting from a smaller model: simply truncate the extraneous columns of $W$.

We construct the task vectors by first collecting a dataset of semantic binaries. These can, for example, be extracted from the analogy dataset. The task vectors are then simply the difference vectors between the embeddings in each binary. In our reported results, we do not normalize either the word embeddings nor the task vectors. However, our robustness checks indicated that choosing to normalize does not qualitatively change our results.

**Fitting the empirical spectra with the Marchenko-Pastur distribution.** The Marchenko-Pastur distribution is a limiting empirical spectral distribution given in terms of an *aspect ratio* $\gamma := N/d$ and an overall scale $\sigma^2$. In particular, if $N < d$ and $\Xi \in \mathbb{R}^{N \times d}$ is a random matrix with i.i.d. elements of mean zero and variance $\sigma^2$, then as $N, d \to \infty$ with $N/d = \gamma$ fixed, the MP law for $d^{-1}\Xi\Xi^\top$ is

$$\mathrm{d}\mu(\lambda) = \frac{1}{2\pi\sigma^2} \frac{\sqrt{(\lambda_+ - \lambda)(\lambda - \lambda_-)}}{\gamma\lambda} \mathrm{d}\lambda \tag{24}$$

$$\lambda_\pm := \sigma^2(1 \pm \sqrt{\gamma})^2 \tag{25}$$

where the support is $\lambda_- \leq \lambda \leq \lambda_+$, and the density is zero outside. Thus, the MP law gives the expected distribution of eigenvalues for a large noisy covariance (or Gram) matrix.

To fit the MP law to the Gram matrix of task vectors, we first subtract off the mean task vector. Then we compute the population variance of the elements of the centered matrix – we use this to set the $\sigma^2$ parameter. We manually fit the aspect ratio $\gamma$, and we find that the best-fitting $\hat{\gamma} = N/d_{\text{eff}}$ corresponds to an effective dimension which is often much lower than the true embedding dimension $d$. We hypothesize that $d_{\text{eff}} < d$ due to anisotropic noise in the task vectors. We report the $d_{\text{eff}}$ resulting from the fit in Figure 4.

**Composition of mean task vector.** We measure the degree to which the spike captures the mean task vector by computing the ratio

$$\text{signal in mean} = \frac{\mathbf{1}^\top R_d R_d^\top \mathbf{1}}{N \cdot \lambda_{\max}(R_d R_d^\top)} \leq 1, \tag{26}$$

where $\mathbf{1}$ is the ones-vector. If this quantity is (close to) 1, then the mean task vector is a large component of the spike, and vice versa. We find that this quantity is consistently greater than 0.9 for all the semantic categories in the analogy benchmark.

# B   Additional figures

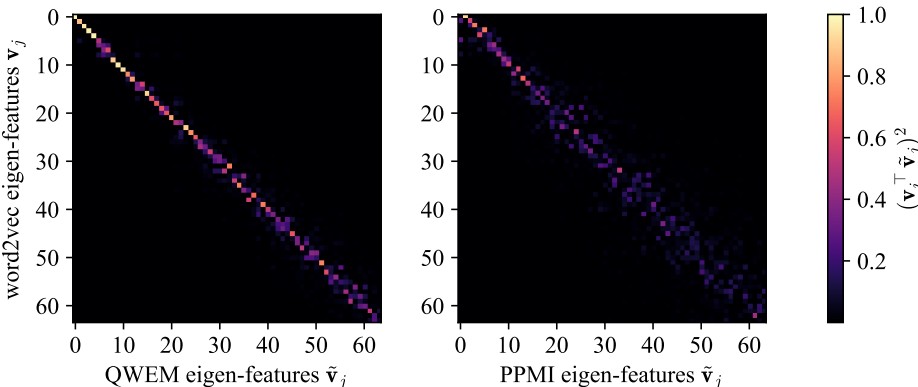

**Figure 5:** `word2vec` **eigen-features align very closely with QWEMs' and less with PPMI.** The heatmap indicates the squared overlaps between the latent feature vectors (i.e., the left singular vectors) between the three models considered. This is a quantitative version of the table in Figure 3.

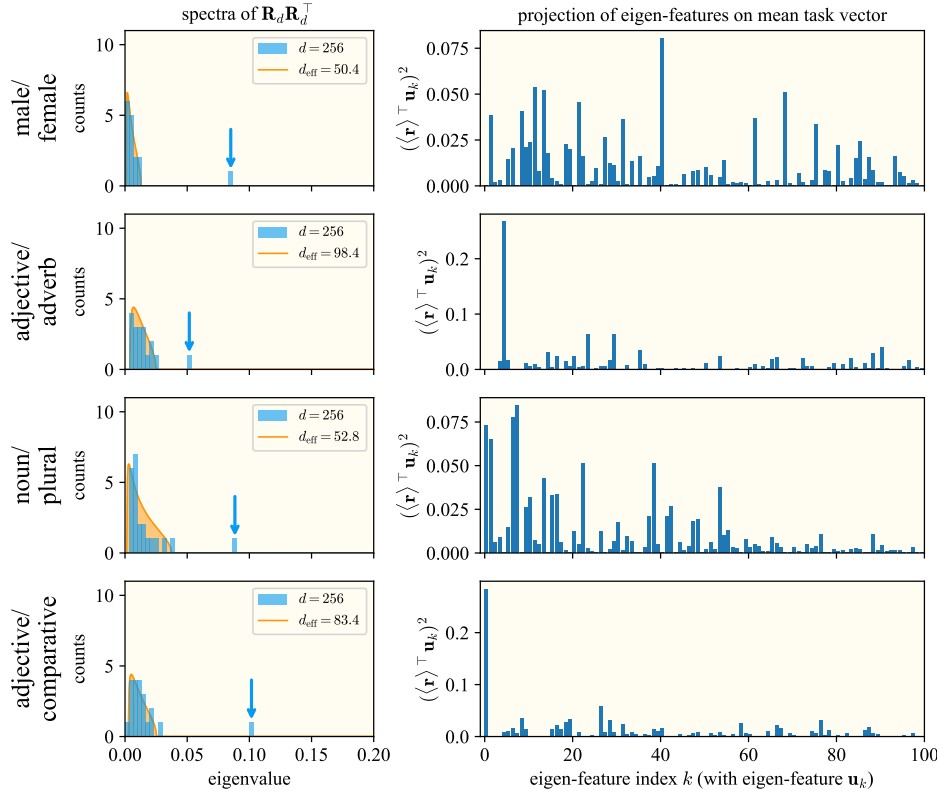

**Figure 6: The empirical observations regarding linear representations extend across the semantic classes in the analogy dataset.** We show that across analogy categories, the corresponding task vectors exhibit the geometric structure discussed in Figure 4.

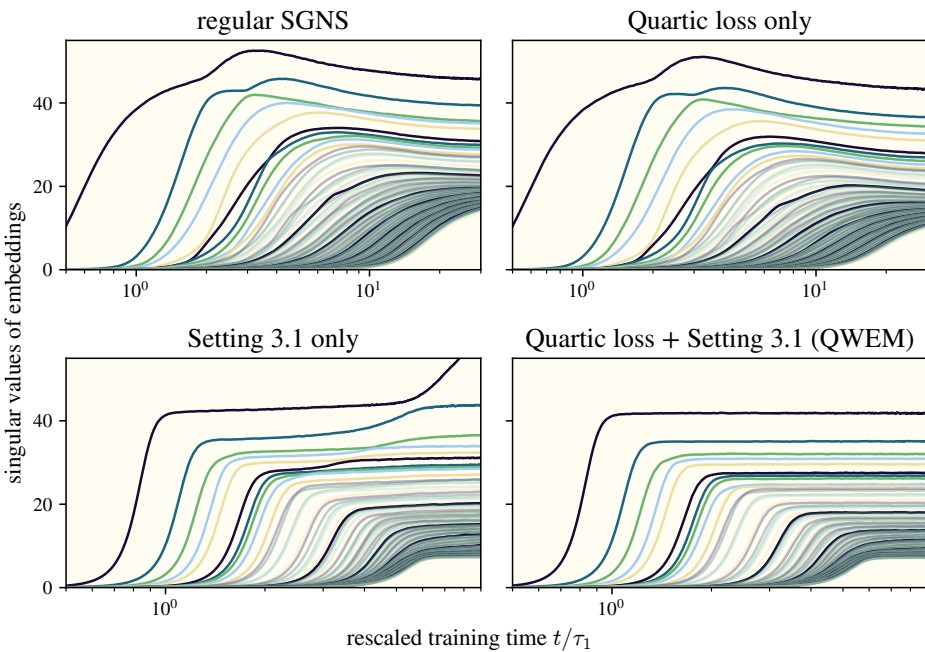

**Figure 7: Ablation tests.** Starting from vanilla `word2vec` (top left), we separately add in both the quartic approximation (top right) and the reweighting condition Setting 3.1 (bottom left). We include QWEM on the bottom right (i.e., both effects). The quartic approximation hardly changes the singular value dynamics at all; the clean mode separation is due to the reweighting. However, as is common with logistic losses, the weights begin to diverge at late times. Using the quartic loss prevents this. The experiments here use a 41-million-token mixture of the Corpus of Contemporary American English and the News on the Web dataset. We use a vocabulary size of 20,000; the embedding dimension is 150; 2 negative samples per positive sample; and a context length of 16.

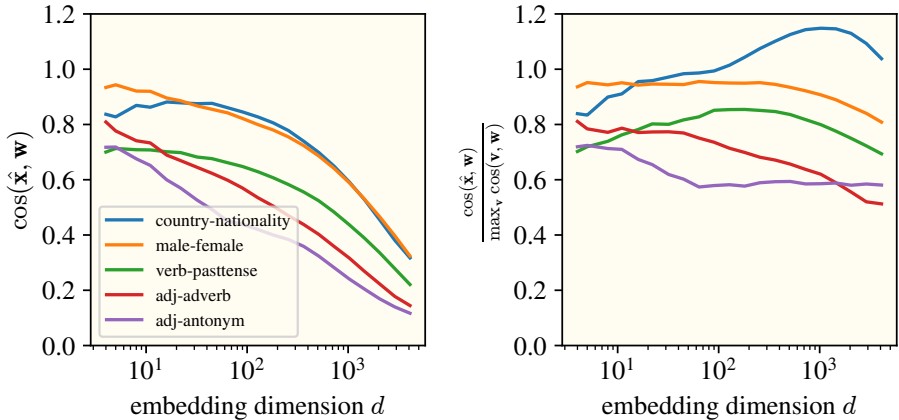

**Figure 8: Although the analogy geometry degrades with $d$, relative scores remain high.** We plot two different measures of analogy geometry across $d$ to illustrate subtleties associated with the top-1 accuracy metric. For an analogy $a : b :: c : w$, we denote $\hat{x} := c + (b - a)$ the prediction for the fourth word. We plot the average cosine similarity between $\hat{x}$ and the true $w$ over the analogy family. The prediction degrades dramatically as the embedding dimension increases, complementing the observation in Figure 4 that the SNR decreases with $d$. However, this geometric breakdown fails to capture the fact that *all* embedding vectors separate as $d$ increases; if we normalize by the maximum cosine similarity between $w$ and non-analogy embeddings, the score remains roughly stable at large $d$. This provides an explanation for why top-1 accuracy often remains high despite the breakdown of geometric analogical structure.

## C Proofs

**Theorem 1** (QWEM = unweighted matrix factorization). *Under Setting 3.1, the contrastive loss Equation (3) can be rewritten as the unweighted matrix factorization problem*

$$\mathcal{L}(\boldsymbol{W}) = \frac{g}{4} \left\| \boldsymbol{W}\boldsymbol{W}^\top - \boldsymbol{M}^\star \right\|_{\mathrm{F}}^2 + \text{const.} \tag{6}$$

*If $\boldsymbol{\Lambda}_{[:d,:d]}$ is positive semidefinite, then the set of global minima of $\mathcal{L}$ is given by*

$$\arg\min_{\boldsymbol{W}} \mathcal{L}(\boldsymbol{W}) = \mathrm{REquiv}\left( \boldsymbol{V}^\star_{[:,:d]} \boldsymbol{\Lambda}^{1/2}_{[:d,:d]} \right). \tag{7}$$

*Proof.* Define $\boldsymbol{M} \coloneqq \boldsymbol{W}\boldsymbol{W}^\top$. Rewriting Equation (3) and plugging in Equation (5) and Setting 3.1,

$$\mathcal{L}(\boldsymbol{W}) = \sum_{i,j} P_{ij}\Psi^+_{ij}\left(\frac{1}{4}\boldsymbol{M}^2_{ij} - \boldsymbol{M}_{ij}\right) + P_iP_j\Psi^-_{ij}\left(\frac{1}{4}\boldsymbol{M}^2_{ij} + \boldsymbol{M}_{ij}\right) \tag{27}$$

$$= \frac{1}{4}\sum_{ij} \boldsymbol{G}_{ij}\boldsymbol{M}^2_{ij} - 4(P_{ij}\Psi^+_{ij} - P_iP_j\Psi^-_{ij})\boldsymbol{M}_{ij} \tag{28}$$

$$= \frac{1}{4}\sum_{ij} \boldsymbol{G}_{ij}\left( \boldsymbol{M}^2_{ij} - 2\frac{P_{ij}\Psi^+_{ij} - P_iP_j\Psi^-_{ij}}{\frac{1}{2}(P_{ij}\Psi^+_{ij} + P_iP_j\Psi^-_{ij})}\boldsymbol{M}_{ij}\right) \tag{29}$$

$$= \frac{1}{4}\sum_{ij} \boldsymbol{G}_{ij}\left( \boldsymbol{M}^2_{ij} - 2\boldsymbol{M}^\star_{ij}\boldsymbol{M}_{ij} + \boldsymbol{M}^{\star\,2}_{ij} - \boldsymbol{M}^{\star\,2}_{ij}\right) \tag{30}$$

$$= \frac{g}{4}\left( \|\boldsymbol{M} - \boldsymbol{M}^\star\|_{\mathrm{F}}^2 + \|\boldsymbol{M}^\star\|_{\mathrm{F}}^2\right). \tag{31}$$

Since $P_{ij}$, $P_iP_j$, $\Psi^+_{ij}$, $\Psi^-_{ij}$ are all real symmetric, so is $\boldsymbol{M}^\star$, so it has an eigendecomposition. By the Eckart-Young-Mirsky theorem, the loss-minimizing $\boldsymbol{M}$ must be the truncated SVD $\boldsymbol{M}^\star_{[d]}$, whose symmetric factors are exactly given by Equation (7). ∎

**Proposition 2.** *Equation (3) can be rewritten as the weighted matrix factorization problem*

$$\mathcal{L}(\boldsymbol{W}) = \frac{1}{4}\sum_{i,j} \boldsymbol{G}_{ij}(\boldsymbol{W}\boldsymbol{W}^\top - \boldsymbol{M}^\star)^2_{ij} + \text{const.} \tag{8}$$

*Let $\Psi^+$ and $\Psi^-$ each be symmetric in $i, j$, let $\boldsymbol{g} \in \mathbb{R}^V$ be an arbitrary vector with non-negative elements, and let $\boldsymbol{\Gamma} = \mathrm{diag}(\boldsymbol{g})^{1/2}$. Then the eigendecomposition of $\boldsymbol{\Gamma}\boldsymbol{M}^\star\boldsymbol{\Gamma} = \boldsymbol{V}^\star_\Gamma\boldsymbol{\Lambda}_\Gamma\boldsymbol{V}^{\star\,\top}_\Gamma$ exists. If $\boldsymbol{G} = \boldsymbol{g}\boldsymbol{g}^\top$ and $\boldsymbol{\Lambda}_{[:d,:d]}$ is positive semidefinite, then the set of global minima of $\mathcal{L}$ is given by*

$$\arg\min_{\boldsymbol{W}} \mathcal{L}(\boldsymbol{W}) = \mathrm{REquiv}\left( \boldsymbol{\Gamma}^{-1}\boldsymbol{V}^\star_{\Gamma,[:,:d]}\boldsymbol{\Lambda}^{1/2}_{\Gamma,[:d,:d]}\right). \tag{9}$$

*Proof.* The formulation as a weighted matrix factorization follows from Equation (30). In the case that $\boldsymbol{G}$ is rank 1, substituting in $\boldsymbol{\Gamma}$, we find

$$\mathcal{L}(\boldsymbol{W}) = \frac{1}{4}\|\boldsymbol{\Gamma}(\boldsymbol{M} - \boldsymbol{M}^\star)\boldsymbol{\Gamma}\|_{\mathrm{F}}^2. \tag{32}$$

After distributing factors and invoking the Eckart-Young-Mirsky theorem, we conclude that the rank-$d$ minimizer is

$$\boldsymbol{M}_{\min} = \boldsymbol{\Gamma}^{-1}\left(\boldsymbol{\Gamma}\boldsymbol{M}^\star\boldsymbol{\Gamma}\right)_{[d]}\boldsymbol{\Gamma}^{-1} \tag{33}$$

whose symmetric factors are exactly given by Equation (9). ∎

**Lemma 3.1** (Training dynamics, aligned initialization). *If* $\tilde{\boldsymbol{\Lambda}} := \boldsymbol{\Lambda}_{[:d,:d]}$ *is positive semidefinite and* $\boldsymbol{V}(0) = \boldsymbol{V}^\star_{[:,:d]}$, *then under Setting 3.1, the gradient flow dynamics* $\frac{\mathrm{d}\boldsymbol{W}}{\mathrm{d}t} = -\frac{1}{2g}\nabla\mathcal{L}(\boldsymbol{W})$ *yields*

$$\boldsymbol{U}(t) = \boldsymbol{U}(0) \tag{10}$$

$$\boldsymbol{S}(t) = \boldsymbol{S}(0)\left(e^{-\tilde{\boldsymbol{\Lambda}}t} + \boldsymbol{S}^2(0)\tilde{\boldsymbol{\Lambda}}^{-1}\left(\boldsymbol{I} - e^{-\tilde{\boldsymbol{\Lambda}}t}\right)\right)^{-1/2} \tag{11}$$

$$\boldsymbol{V}(t) = \boldsymbol{V}(0) = \boldsymbol{V}^\star_{[:,:d]} \tag{12}$$

*Proof.* By Theorem 1, we write the loss (Equation (3)) as

$$\mathcal{L}(\boldsymbol{W}) = \frac{g}{4}\operatorname{Tr}\left[(\boldsymbol{W}\boldsymbol{W}^\top - \boldsymbol{M}^\star)^\top(\boldsymbol{W}\boldsymbol{W}^\top - \boldsymbol{M}^\star)\right]. \tag{34}$$

We neglect constant terms since they do not affect the gradient descent dynamics. Under the stated gradient flow, the equation of motion is

$$\frac{\mathrm{d}\boldsymbol{W}}{\mathrm{d}t} = -\frac{1}{2g}\nabla\mathcal{L}(\boldsymbol{W}) \tag{35}$$

$$= -\frac{1}{8}\left(2(\boldsymbol{W}\boldsymbol{W}^\top - \boldsymbol{M}^\star)(2\boldsymbol{W})\right) \tag{36}$$

$$= \frac{1}{2}(\boldsymbol{M}^\star - \boldsymbol{W}\boldsymbol{W}^\top)\boldsymbol{W}. \tag{37}$$

From here on we adopt the dot notation for time derivatives. We use the spectral decompositions $\boldsymbol{W}(t)^\top = \boldsymbol{U}(t)\boldsymbol{S}(t)\boldsymbol{V}(t)^\top$ and $\boldsymbol{M}^\star = \boldsymbol{V}^\star\boldsymbol{\Lambda}\boldsymbol{V}^{\star\top}$. Then the above can be written

$$\dot{\boldsymbol{W}} = \dot{\boldsymbol{V}}\boldsymbol{S}\boldsymbol{U}^\top + \boldsymbol{V}\dot{\boldsymbol{S}}\boldsymbol{U}^\top + \boldsymbol{V}\boldsymbol{S}\dot{\boldsymbol{U}}^\top = \frac{1}{2}(\boldsymbol{V}^\star\boldsymbol{\Lambda}\boldsymbol{V}^{\star\top}\boldsymbol{V} - \boldsymbol{V}\boldsymbol{S}^2)\boldsymbol{S}\boldsymbol{U}^\top. \tag{38}$$

Left-multiplying by $\boldsymbol{V}^\top$ and right-multiplying by $\boldsymbol{U}$, we obtain

$$\boldsymbol{V}^\top\dot{\boldsymbol{V}}\boldsymbol{S} + \dot{\boldsymbol{S}} + \boldsymbol{S}\dot{\boldsymbol{U}}^\top\boldsymbol{U} = \frac{1}{2}(\tilde{\boldsymbol{\Lambda}} - \boldsymbol{S}^2)\boldsymbol{S}. \tag{39}$$

where we now use the alignment assumption, $\boldsymbol{V}^\top\boldsymbol{V}^\star = \boldsymbol{I}$. Note that since we use the economy SVD in our notation, we use a non-standard notation where $\boldsymbol{I} \in \mathbb{R}^{d\times V}$ is a rectangular matrix with ones on the main diagonal and zeros elsewhere. Now, since the RHS is diagonal, we must have that

$$(\boldsymbol{V}^\top\dot{\boldsymbol{V}}\boldsymbol{S} + \boldsymbol{S}\dot{\boldsymbol{U}}^\top\boldsymbol{U})_{ij} = 0 \qquad \text{for} \qquad i \neq j. \tag{40}$$

Furthermore, since $\boldsymbol{V}$ and $\boldsymbol{U}$ are orthogonal, $\boldsymbol{V}^\top\dot{\boldsymbol{V}}$ and $\dot{\boldsymbol{U}}^\top\boldsymbol{U}$ must both be antisymmetric (since for any orthogonal matrix $\boldsymbol{Q}$, $\frac{\mathrm{d}}{\mathrm{d}t}\boldsymbol{Q}^\top\boldsymbol{Q} = \dot{\boldsymbol{Q}}^\top\boldsymbol{Q} + \boldsymbol{Q}^\top\dot{\boldsymbol{Q}} = \dot{\boldsymbol{I}} = 0$). It follows that, for all $i \neq j$,

$$(\boldsymbol{V}^\top\dot{\boldsymbol{V}})_{ij}s_i + (\dot{\boldsymbol{U}}^\top\boldsymbol{U})_{ij}s_j = (\boldsymbol{V}^\top\dot{\boldsymbol{V}})_{ij}s_j + (\dot{\boldsymbol{U}}^\top\boldsymbol{U})_{ij}s_i = 0. \tag{41}$$

Isolating $(\boldsymbol{V}^\top\dot{\boldsymbol{V}})_{ij}$, we see that this can only hold if $s_i = s_j$ or if $(\boldsymbol{V}^\top\dot{\boldsymbol{V}})_{ij} = (\dot{\boldsymbol{U}}^\top\boldsymbol{U})_{ij} = 0$. The former is ruled out by level repulsion in Gaussian random matrices; with probability 1 we have that $\boldsymbol{S}$ contains distinct singular values. We conclude that $\dot{\boldsymbol{U}} = 0$ and $\dot{\boldsymbol{V}} = 0$.

Returning to Equation (39), we have that

$$\dot{\boldsymbol{S}} = \frac{1}{2}(\tilde{\boldsymbol{\Lambda}} - \boldsymbol{S}^2)\boldsymbol{S}. \tag{42}$$

These are precisely the dynamics studied in Saxe et al. (2014). These dynamics are now decoupled, so we may solve each component separately. The solution to this equation is

$$s_k^2(t) = \frac{s_k^2(0)\,\lambda_k\,e^{\lambda_k t}}{\lambda_k + s_k^2(0)\,(e^{\lambda_k t} - 1)}. \tag{43}$$

Thus, the each singular direction of the embeddings is realized in a characteristic time

$$\tau_k = \frac{1}{\lambda_k}\ln\frac{\lambda_k}{s_k^2(0)}. \qquad \blacksquare \tag{44}$$

**Result 3** (Training dynamics, vanishing random initialization). *Initialize $\boldsymbol{W}_{ij}(0) \sim \mathcal{N}(0, \sigma^2)$, and let $\boldsymbol{S}(0)$ denote the singular values of $\boldsymbol{W}(0)$. Define the characteristic time $\tau_1 := \lambda_1^{-1} \ln(\lambda_1/\sigma^2)$ and the rescaled time variable $\tilde{t} = t/\tau_1$. Define $\boldsymbol{W}^\star(0) := \boldsymbol{V}^\star_{[:,:d]} \boldsymbol{S}(0)$ and let $\boldsymbol{W}^\star(t)$ be its gradient flow trajectory given by Lemma 3.1. If $\boldsymbol{\Lambda}_{[:d,:d]}$ is positive semidefinite, then under Setting 3.1 we have with high probability that*

$$\forall \tilde{t} \geq 0, \quad \lim_{\sigma^2 \to 0} \min_{\boldsymbol{W}' \in \mathrm{REquiv}(\boldsymbol{W}(\tilde{t}\tau_1))} \|\boldsymbol{W}' - \boldsymbol{W}^\star(\tilde{t}\tau_1)\|_{\mathrm{F}} = 0. \tag{13}$$

*Derivation.* Before starting the main derivation, we give a qualitative argument for why one expects the result of Lemma 3.1 to hold in the small initialization limit despite a lack of initial alignment.

**Warmup.** We begin with the equation of motion for $\boldsymbol{M} := \boldsymbol{W}\boldsymbol{W}^\top$:

$$\dot{\boldsymbol{M}} = \boldsymbol{W}\dot{\boldsymbol{W}}^\top + \dot{\boldsymbol{W}}\boldsymbol{W}^\top = \frac{1}{2}\left(\boldsymbol{M}\boldsymbol{M}^\star + \boldsymbol{M}^\star\boldsymbol{M} - 2\boldsymbol{M}^2\right). \tag{45}$$

We again consider the dynamics in terms of the spectral decompositions $\boldsymbol{M}(t) = \boldsymbol{V}(t)\boldsymbol{S}^2(t)\boldsymbol{V}(t)^\top$ and $\boldsymbol{M}^\star = \boldsymbol{V}^\star\boldsymbol{\Lambda}^\star\boldsymbol{V}^{\star\top}$. Note that here $\boldsymbol{V}, \boldsymbol{S} \in \mathbb{R}^{V \times V}$ are square. We define the eigenbasis overlap $\boldsymbol{O} := \boldsymbol{V}^{\star\top}\boldsymbol{V}$. After transforming coordinates to the target eigenbasis, we find

$$\boldsymbol{V}^{\star\top}\dot{\boldsymbol{M}}\boldsymbol{V}^\star = \boldsymbol{V}^{\star\top}(\dot{\boldsymbol{V}}\boldsymbol{S}^2\boldsymbol{V}^\top + \boldsymbol{V}(2\boldsymbol{S}\dot{\boldsymbol{S}})\boldsymbol{V}^\top + \boldsymbol{V}\boldsymbol{S}^2\dot{\boldsymbol{V}}^\top)\boldsymbol{V}^\star \tag{46}$$

$$= \dot{\boldsymbol{O}}\boldsymbol{S}^2\boldsymbol{O}^\top + 2\boldsymbol{O}\boldsymbol{S}\dot{\boldsymbol{S}}\boldsymbol{O}^\top + \boldsymbol{O}\boldsymbol{S}^2\dot{\boldsymbol{O}}^\top \tag{47}$$

$$= \frac{\boldsymbol{\Lambda}\boldsymbol{O}\boldsymbol{S}^2\boldsymbol{O}^\top + \boldsymbol{O}\boldsymbol{S}^2\boldsymbol{O}^\top\boldsymbol{\Lambda}}{2} - \boldsymbol{O}\boldsymbol{S}^4\boldsymbol{O}^\top. \tag{48}$$

For clarity, we rotate coordinates again into the $\boldsymbol{O}$ basis and find

$$\boldsymbol{S}^2\dot{\boldsymbol{O}}^\top\boldsymbol{O} + \boldsymbol{O}^\top\dot{\boldsymbol{O}}\boldsymbol{S}^2 + 2\boldsymbol{S}\dot{\boldsymbol{S}} = \frac{\boldsymbol{S}^2\boldsymbol{O}^\top\boldsymbol{\Lambda}\boldsymbol{O} + \boldsymbol{O}^\top\boldsymbol{\Lambda}\boldsymbol{O}\boldsymbol{S}^2}{2} - \boldsymbol{S}^4. \tag{49}$$

Since $\boldsymbol{O}$ is orthogonal, it satisfies $\dot{\boldsymbol{O}}^\top\boldsymbol{O} + \boldsymbol{O}^\top\dot{\boldsymbol{O}} = \boldsymbol{0}$ (this follows from differentiating the identity $\boldsymbol{O}^\top\boldsymbol{O} = \boldsymbol{I}$). Therefore the first two terms on the LHS of Equation (49), which concern the singular vector dynamics, have zero diagonal; the third term, which concerns singular value dynamics, has zero off-diagonal. This implies

$$2\boldsymbol{S}\dot{\boldsymbol{S}} = \left(\mathrm{diag}(\boldsymbol{O}^\top\boldsymbol{\Lambda}\boldsymbol{O}) - \boldsymbol{S}^2\right)\boldsymbol{S}^2, \tag{50}$$

where $\mathrm{diag}(\cdot)$ is the diagonal matrix formed from the diagonal of the argument. From Equation (50), we see that at initialization $2\boldsymbol{S}\dot{\boldsymbol{S}}$ scales with the initialization size $\sigma^2$ since $\boldsymbol{S}^2(0) \sim \sigma^2$. On the other hand, from the off-diagonal of Equation (49), we see that $\dot{\boldsymbol{O}}$ is order 1 (since the scale of $\boldsymbol{O}$ is fixed by orthonormality). Therefore, in the limit of small initialization, we expect the model to align quickly compared to the dynamics of $\boldsymbol{S}^2$. This motivates the *silent alignment ansatz*, which informally posits that with high probability, the top $d \times d$ submatrix of $\boldsymbol{O}$ converges towards the identity matrix well before $\boldsymbol{S}^2$ reaches the scale of $\boldsymbol{\Lambda}$. As $\boldsymbol{O} \to \boldsymbol{I}$, the dynamics decouple and enter a regime well-described by Lemma 3.1. We formalize this idea in our concrete derivation of Result 3.

**Main derivation.** We are interested in showing that

$$\forall \tilde{t} \geq 0, \quad \lim_{\sigma^2 \to 0} \min_{\boldsymbol{W}' \in \mathrm{REquiv}(\boldsymbol{W}(\tilde{t}\tau_1))} \|\boldsymbol{W}' - \boldsymbol{W}^\star(\tilde{t}\tau_1)\|_{\mathrm{F}} = 0, \tag{51}$$

where we express the statement in terms of rescaled time $\tilde{t} = t/\tau_1$ since we anticipate that $\tau_1 \to \infty$ as $\sigma^2 \to 0$. For notational clarity, though, let us switch back to the original time variable. Then

$$\min_{\boldsymbol{W}'(t) \in \mathrm{REquiv}(\boldsymbol{W}(t))} \|\boldsymbol{W}'(t) - \boldsymbol{W}^\star(t)\|_{\mathrm{F}} = \min_{\boldsymbol{W}'(t) \in \mathrm{REquiv}(\boldsymbol{W}(t))} \|\boldsymbol{W}'(t)^\top - \boldsymbol{W}^\star(t)^\top\|_{\mathrm{F}} \tag{52}$$

$$= \min_{\boldsymbol{U}'(t) \in \mathcal{O}} \|\boldsymbol{U}'(t)\boldsymbol{U}(t)\boldsymbol{S}(t)\boldsymbol{V}^\top(t) - \boldsymbol{S}^\star(t)\boldsymbol{V}^{\star\top}\|_{\mathrm{F}} \tag{53}$$

$$= \min_{\boldsymbol{U}'(t) \in \mathcal{O}} \|\boldsymbol{U}'(t)\boldsymbol{U}(t)\boldsymbol{S}(t)\boldsymbol{O}^\top(t) - \boldsymbol{S}^\star(t)\|_{\mathrm{F}} \tag{54}$$

where we again define the eigenbasis overlap $\boldsymbol{O} := \boldsymbol{V}^{\star\top}\boldsymbol{V}$.

Motivated by the expectation that we will see sequential learning dynamics starting from the top mode and descending into lower modes, we seek a change of variables in which the dynamics are expressed in an upper-triangular matrix. We can achieve this reparameterization using a QR factorization: $USO^\top \to QR$, where $Q$ is orthogonal and $R$ is upper triangular. Then

$$\min_{W'(t)\in\text{REquiv}(W(t))} \|W'(t) - W^\star(t)\|_\text{F} = \min_{U'(t)\in\mathcal{O}} \|U'(t)Q(t)R(t) - S^\star(t)\|_\text{F} \qquad (55)$$

$$= \min_{U'(t)\in\mathcal{O}} \|U'(t)R(t) - S^\star(t)\|_\text{F}, \qquad (56)$$

where the second equation follows since $U'$ is a variable to be minimized, and it can simply absorb $Q$. We emphasize that this convenience follows from the right orthogonal symmetry of the embeddings. If we can show that $\|R(t) - S^\star(t)\|_\text{F} \to 0$ for all $t$ in the vanishing initialization limit, then we will have completed the derivation.

Our starting point will be transpose of Equation (37), right-multiplied by $V^\star$:

$$\frac{\text{d}}{\text{d}t}(USO^\top) = \frac{1}{2}USO^\top(\Lambda - OS^2O^\top) \implies \dot{Q}R + Q\dot{R} = \frac{1}{2}QR\left(\Lambda - R^\top R\right). \qquad (57)$$

Left-multiplying by $2Q^\top$ and rearranging, we find

$$2\dot{R} = R\left(\Lambda - R^\top R\right) - 2Q^\top \dot{Q}R \qquad (58)$$

$$= R\Lambda - (RR^\top + 2Q^\top \dot{Q})R \qquad (59)$$

$$= R\Lambda - \tilde{R}R, \qquad (60)$$

where we define $\tilde{R} := RR^\top + 2Q^\top \dot{Q}$. Note that $\tilde{R}$ must be upper triangular: since the other terms in the equation are upper triangular, so must be $\tilde{R}R$; and since $R$ and $\tilde{R}R$ are both upper triangular, then $\tilde{R}$ must be upper triangular.

In fact, this is enough to fully determine the elements of $\tilde{R}$. We know that $Q^\top \dot{Q}$ is antisymmetric (since $Q^\top Q = I$ by orthogonality, $Q^\top \dot{Q} + \dot{Q}^\top Q = 0$). Additionally using the fact that $RR^\top$ is symmetric and imposing upper-triangularity on the sum, we have that

$$\tilde{R}_{ij} = \begin{cases} 2(RR^\top)_{ij} & \text{if} \quad i < j \\ (RR^\top)_{ii} & \text{if} \quad i = j \\ 0 & \text{if} \quad i > j \end{cases}. \qquad (61)$$

Here, we take a moment to examine the dynamics in Equation (60). Treating the initialization scale $\sigma$ as a scaling variable, we expect that $R_{ij} \sim \sigma$. Thus, in the small initialization limit, we expect the second term (which scales like $\sigma^3$) to contribute negligibly until late times; initially, we will see an exponential growth in the elements of $R$ with growth rates given by $\Lambda$. Later, $R$ will (roughly speaking) reach the scale of $\Lambda^{1/2}$, and there will be competitive dynamics between the two terms. We now write out the element-wise dynamics of $R$ to see this precisely.

$$2\dot{R}_{ij} = R_{ij}\lambda_j - \sum_{j\geq k\geq i} \tilde{R}_{ik}R_{kj} \qquad (62)$$

$$= R_{ij}\lambda_j - \sum_{j\geq k\geq i}\sum_{\ell\geq k}(2 - \delta_{ik})R_{i\ell}R_{k\ell}R_{kj} \qquad (63)$$

$$= R_{ij}\lambda_j - \sum_{\ell\geq i}R_{i\ell}^2R_{ij} - 2\sum_{j\geq k>i}\sum_{\ell\geq k}R_{i\ell}R_{k\ell}R_{kj} \qquad (64)$$

$$= R_{ij}\lambda_j - \sum_{\ell\geq i}R_{i\ell}^2R_{ij} - 2\sum_{j\geq k>i}R_{ij}R_{kj}^2 - 2\sum_{j\geq k>i}\sum_{j>\ell\geq k}R_{i\ell}R_{k\ell}R_{kj} \qquad (65)$$

$$= \left(\lambda_j - \sum_{\ell\geq i}R_{i\ell}^2 - 2\sum_{j\geq k>i}R_{kj}^2\right)R_{ij} - 2\sum_{j\geq k>i}\sum_{j>\ell\geq k}R_{i\ell}R_{k\ell}R_{kj}. \qquad (66)$$

We have separated the dynamics of $R_{ij}$ into a part that is explicitly linear in $R_{ij}$ and a part which has no explicit dependence on $R_{ij}$. (Of course, there is coupling between all the elements of $R$ and $R_{ij}$ through their own dynamical equations.) So far, everything we have done is exact.

Our main approximation is to argue that at all times, only the diagonal elements of $\boldsymbol{R}$ contribute non-negligibly to the dynamics. This holds if the off-diagonal elements are initialized vanishingly small and if they remain vanishingly small throughout. In this case, we may discard any terms that include couplings between off-diagonal elements.

With this approximation, we may discard the entire second term on the RHS, as well as some of the summands in the first prefactor for $\boldsymbol{R}_{ij}$. This leaves the approximate dynamics

$$2\dot{\boldsymbol{R}}_{ij} = \left(\lambda_j - \boldsymbol{R}_{ii}^2 - 2(1-\delta_{ij})\boldsymbol{R}_{jj}^2\right)\boldsymbol{R}_{ij}. \tag{67}$$

We may now directly solve for the diagonal dynamics. Letting $r_k := \boldsymbol{R}_{kk}$,

$$\dot{r}_k = \frac{1}{2}\left(\lambda_k - r_k^2\right)r_k \quad \implies \quad r_k^2(t) = \frac{r_k^2(0)\,\lambda_k\,e^{\lambda_k t}}{\lambda_k + r_k^2(0)\,(e^{\lambda_k t}-1)}. \tag{68}$$

We obtain the same sigmoidal dynamics as in Lemma 3.1. If we show that the off-diagonal elements remain vanishingly small in the limit $\sigma^2 \to 0$, then: a) our approximation is justified, and b) it follows that $\|\boldsymbol{R}(t) - \boldsymbol{S}^\star(t)\|_{\mathrm{F}} \to 0$ for all $t$, completing the derivation.

To do this, we examine the dynamics of the off-diagonals and show that the maximum scale they achieve (at any time) decays to zero as $\sigma^2 \to 0$. For $i < j$ we have

$$2\dot{\boldsymbol{R}}_{ij} = \left(\lambda_j - r_i^2 - 2r_j^2\right)\boldsymbol{R}_{ij} \tag{69}$$

This is a linear first-order homogeneous ODE with a time-dependent coefficient, and thus it can be solved exactly:

$$\boldsymbol{R}_{ij}^2(t) = \boldsymbol{R}_{ij}^2(0)\,\cdot\left(\frac{\lambda_j}{\lambda_j + r_j^2(0)\,(e^{\lambda_j t}-1)}\right)^2 \cdot \frac{\lambda_i}{\lambda_i + r_i^2(0)\,(e^{\lambda_i t}-1)} \cdot e^{\lambda_j t} \tag{70}$$

$$= \boldsymbol{R}_{ij}^2(0) \cdot \frac{r_j^4(t)}{r_j^4(0)} \cdot \frac{r_i^2(t)}{r_i^2(0)} \cdot e^{-(\lambda_i + \lambda_j)t}. \tag{71}$$

This product contains two factors with sigmoidal dynamics of different timescales, and one factor with an exponential decay to the dynamics. Thus, as $t \to \infty$, the first two factors saturate, and the decay drives the off-diagonal elements $\boldsymbol{R}_{ij}$ to zero. We now show that $\max_t \boldsymbol{R}_{ij}^2(t)$ vanishes as $\sigma^2 \to 0$. Focusing on the scaling w.r.t. the initialization scale, we may approximate $\boldsymbol{R}_{k\ell}^2(0) \approx \sigma^2$ for all $k \leq \ell$. Discarding $O(\sigma^4)$ terms and solving $\dot{\boldsymbol{R}}_{ij} = 0$, we find

$$\max_t \boldsymbol{R}_{ij}^2 \approx \left(\frac{\lambda_i \lambda_j}{\lambda_i - \lambda_j}\right)^{\lambda_j/\lambda_i}\sigma^{2(\lambda_i - \lambda_j)/\lambda_i} \quad \text{when} \quad t = \frac{1}{\lambda_i}\log\frac{\lambda_i \lambda_j}{\sigma^2(\lambda_i - \lambda_j)}. \tag{72}$$

Therefore, for $i < j$, assuming $\lambda_i \neq \lambda_j$,

$$\lim_{\sigma^2 \to 0}\max_t \boldsymbol{R}_{ij}^2 = 0 \tag{73}$$

and we conclude that $\|\boldsymbol{R}(t) - \boldsymbol{S}^\star(t)\|_{\mathrm{F}} \to 0$ as desired.

In fact, under these approximations, as long as the initialization scale satisfies

$$\log\sigma^2 \ll -\frac{\lambda_j}{\lambda_i - \lambda_j}\log\left(\frac{\lambda_i \lambda_j}{\lambda_i - \lambda_j}\right) \tag{74}$$

for all $i$ and $j$, the off-diagonal elements will remain much smaller than the diagonal elements, and we may view the diagonal element dynamics as simply being the singular value dynamics. This follows from Weyl's inequality for matrix perturbations. Thus we may expect that our results hold in the small-but-finite initialization regime (e.g., the regime accessed by our experiments). ∎

# D  Relation between QWEMs and known algorithms

## D.1  Relation to PMI

Early word embedding algorithms obtained low-dimensional embeddings by explicitly constructing some target matrix and employing a dimensionality reduction algorithm. One popular choice was the *pointwise mutual information* (PMI) matrix (Church & Hanks, 1990), defined

$$\text{PMI}_{ij} = \log \frac{P_{ij}}{P_i P_j}. \tag{75}$$

Later, Levy & Goldberg (2014) showed that PMI is the rank-unconstrained minimizer of $\mathcal{L}_{\texttt{w2v}}$. To see the relation between the QWEM target $\boldsymbol{M}^\star$ and PMI, let us write

$$\frac{P_{ij}}{P_i P_j} = 1 + \Delta(x_{ij}), \tag{76}$$

where the function $\Delta(x)$ yields the fractional deviation away from independent word statistics, in terms of some small parameter $x$ of our choosing (so that $\Delta(0) = 0$). This setup allows us to Taylor expand quantities of interest around $x = 0$. A judicious choice will produce terms that cancel the $-\frac{1}{2}\Delta^2$ that arises from the Taylor expansion of $\log(1 + \Delta)$, leaving only third-order corrections. One such example is $\Delta(x) = 2x/(2 - x)$, which yields

$$x_{ij} = \frac{P_{ij} - P_i P_j}{\frac{1}{2}(P_{ij} + P_i P_j)} = \boldsymbol{M}^\star_{ij} \tag{77}$$

and

$$\text{PMI} = \log\left(1 + \frac{2x}{2 - x}\right) = x + \frac{x^3}{12} + \frac{x^5}{80} + \cdots \tag{78}$$

This calculation reveals that $\boldsymbol{M}^\star$ learns a very close approximation to the PMI matrix; the leading correction is third order. However, $\boldsymbol{M}^\star$ is much friendlier to least-squares approximation, since $x$ is bounded ($-2 \leq \boldsymbol{M}^\star_{ij} \leq 2$).

## D.2  Relation to SimCLR

SimCLR is a widely-used contrastive learning algorithm for learning visual representations (Chen et al., 2020). It uses a deep convolutional encoder to produce latent representations from input images. Data augmentation is used to construct positive pairs; negative pairs are drawn uniformly from the dataset. The encoder is then trained using the *normalized temperature-scaled cross entropy loss*:

$$\mathcal{L}(\boldsymbol{f}_\theta) = \underset{i,j \sim \Pr(\cdot,\cdot)}{\mathbb{E}} \left[ -\log \frac{\exp\left(\beta \boldsymbol{f}_\theta(x_i)^\top \boldsymbol{f}_\theta(x_j)\right)}{\sum_{k \neq j}^{B} \exp\left(\beta \boldsymbol{f}_\theta(x_i)^\top \boldsymbol{f}_\theta(x_k)\right)} \right], \tag{79}$$

where $\Pr(\cdot, \cdot)$ is the positive pair distribution, $\boldsymbol{f}_\theta x_i$ is the learned representation of $x_i$, $\beta$ is an inverse temperature hyperparameter, and $B$ is the batch size. Defining $\boldsymbol{S}_\theta(i,j) = \boldsymbol{f}_\theta(x_i)^\top \boldsymbol{f}_\theta(x_j)$, in the limit of large batch size, we can Taylor expand this objective function around the origin:

$$\mathcal{L}(\boldsymbol{S}_\theta) = \underset{i,j \sim \Pr(\cdot,\cdot)}{\mathbb{E}} \left[ -\beta \boldsymbol{S}_\theta(i,j) + \log\left( \underset{k \sim \Pr(\cdot)}{\mathbb{E}} \left[ \exp(\beta \boldsymbol{S}_\theta(i,k)) \right] \right) + \log B \right] \tag{80}$$

$$\approx \underset{i,j \sim \Pr(\cdot,\cdot)}{\mathbb{E}} \left[ -\beta \boldsymbol{S}_\theta(i,j) + \underset{k \sim \Pr(\cdot)}{\mathbb{E}} \left[ \exp(\beta \boldsymbol{S}_\theta(i,k)) \right] - 1 \right] + \log B \tag{81}$$

$$\approx \underset{i,j \sim \Pr(\cdot,\cdot)}{\mathbb{E}} \left[ -\beta \boldsymbol{S}_\theta(i,j) \right] + \underset{\substack{i \sim \Pr(\cdot) \\ k \sim \Pr(\cdot)}}{\mathbb{E}} \left[ 1 + \beta \boldsymbol{S}_\theta(i,k) + \frac{1}{2}\beta^2 \boldsymbol{S}_\theta^2(i,k) \right] - 1 + \log B \tag{82}$$

$$\approx \beta \left( \underset{i,j \sim \Pr(\cdot,\cdot)}{\mathbb{E}} \left[ -\boldsymbol{S}_\theta(i,j) \right] + \underset{\substack{i \sim \Pr(\cdot) \\ j \sim \Pr(\cdot)}}{\mathbb{E}} \left[ \boldsymbol{S}_\theta(i,j) + \frac{\beta}{2} \boldsymbol{S}_\theta^2(i,j) \right] \right) + \text{const.} \tag{83}$$

$$\tag{84}$$

If the model is in a linearized regime, we may approximate $S_\theta(i,j) \approx \boldsymbol{f}_i^\top \boldsymbol{f}_j$ for some linearized feature vectors $\boldsymbol{f}$. Then the loss can be written as an unweighted matrix factorization problem using exactly the same argument as in Theorem 1. Thus, we expect that vision models trained under the SimCLR loss in a linearized regime will undergo stepwise sigmoidal dynamics. This provides an explanation for the previously unresolved observation in Simon et al. (2023b) that vision models trained with SimCLR from small initialization exhibit stepwise learning.

### D.3 Relation to next-token prediction.

Word embedding targets are order-2 tensors $\boldsymbol{M}^\star$ that captures two-token (skip-gram) statistics. These two-token statistics are sufficient for coarse semantic understanding tasks such as analogy completion. To perform well on more sophisticated tasks, however, requires modeling more sophisticated language distributions.

The current LLM paradigm demonstrates that the next-token distribution is largely sufficient for most downstream tasks of interest. The next-token prediction (NTP) task aims to model the probability of finding word $i$ given a preceding window of context tokens of length $L-1$. Therefore, the NTP target is an order-$L$ tensor that captures the joint distribution of length-$L$ contexts. NTP thus *generalizes* the word embedding task. Both QWEM and LLMs are underparameterized models that learn internal representations with interpretable and task-relevant vector structure. Both are trained using self-supervised gradient descent algorithms, *implicitly* learning a compression of natural language statistics by iterating through the corpus.

Although the size of the NTP solution space is exponential in $L$ (i.e., much larger than that of QWEM), LLMs succeed because the sparsity of the target tensor increases with $L$. We conjecture, then, that a dynamical description of learning sparse high-dimensional tensors is necessary for a general scientific theory of when and how LLMs succeed on reasoning tasks and exhibit failures such as hallucinations or prompt attack vulnerabilities.

### D.4 Relation to neural tangent kernel.

Our result describing an implicit bias towards low rank directly contrasts the well-studied neural tangent kernel training (NTK) regime (Jacot et al., 2018; Chizat et al., 2019; Karkada, 2024). Here we compare the two learning regimes.

**Learning with NTK.**

- Learning dynamics and generalization performance can be solved (Lee et al., 2019; Bordelon et al., 2020; Simon et al., 2023a).
- Extreme over-parameterization is required; large finite-width corrections at practical widths (Huang & Yau, 2020).
- Model weights do not learn task-relevant features.
- Optimization remains in a locally convex region of the loss landscape (Chizat et al., 2019).

**Learning from small initialization.**

- Learning dynamics are generally complicated; can be solved in very simple cases (linear networks, special data distributions, etc.).
- Behavior is qualitatively consistent across network widths, with only moderate finite-width corrections (Vyas et al., 2023).
- Model weights learn task-relevant features.
- Optimization tends to pass near a sequence of saddle points. (Baldi & Hornik, 1989; Jacot et al., 2021).

