# OpenReview forum: "Closed-Form Training Dynamics Reveal Learned Features and Linear Structure in Word2Vec-like Models"
_NeurIPS.cc/2025/Conference — NeurIPS 2025 poster_

### Official Review · Reviewer_TGpe · 2025-07-01

**Clarity:** 3
**Significance:** 2
**Originality:** 3
**Rating:** 4
**Confidence:** 3

**Summary:**

This paper provides an analytical study of self-supervised word embedding learning by introducing Quadratic Word Embedding Models (QWEM), derived via a quartic Taylor approximation of the word2vec loss function. The authors analyze both the training dynamics and the final solution under gradient flow, revealing a rank-incremental learning process that matches empirical observations in standard word2vec training. They demonstrate that QWEM closely approximates word2vec in terms of learned features and benchmark performance, while being analytically tractable. The study also explores the emergence of linear representations in latent space and proposes a random matrix theory perspective on task vectors in analogy tasks.

**Questions:**

Is there any possible relations between your focus and some more modern methods?

Is this possible to give a more explicit explanation of why gradient descent in Word2vec leads to a reasonable embedding(or similar to QWEM rather than directly approximating PMI)?

In figure 4, SNR and task performance don't really match, and some tasks suggest that bigger d , better performance. I think this part is not really convincing.

**Ethical Concerns:**

["NO or VERY MINOR ethics concerns only"]

**Final Justification:**

Based on the authors’ rebuttal and the strengths I mentioned in my initial review, I have decided to keep my score.

**Limitations:**

Yes

**Quality:**

3

**Strengths And Weaknesses:**

The story of theory part is clear, and authors are indeed trying to provide experimental evidence showing that theory matches practice. The idea of taylor approximating the loss function is simple but seemingly effective of understanding behaviors of word2vec. Introducing M, which is element-wise bounded and hence able to solve the inherent problem of PMI, shows a strong ability to capture the concrete matrix factorization which word2vec learns.

However, word2vec is simple and far from SOTA models that could be very complex, leading to only limited insight into understanding the success of recent LLMs. As a result, the significance and utility of this paper could be somehow limited. Also, there is no theoretical guarantee that QWEM would always behave similarly to original word2vec. Though the paper claims to provide a closed-form solution of training dynamics, this solution is under a lot of assumptions unverified or only verified by single evidence. The similarity of behaviors could be dependent on datasets and models with different sizes and features, which is partially admitted in the paper.

---

> ### Author Rebuttal · Authors · 2025-07-30
>
> Thank you for your assessment and review, we appreciate it. We are glad that you found our results clear and our main idea effective. Some comments regarding your concerns and questions:
>
> **What’s the point of this analysis? Isn’t word2vec outdated?** Actually, the core mechanistic idea behind word2vec is back in vogue! Many recent papers empirically report linear structure in the latent representations in LLMs – a direct analog of the word2vec embedding geometry. (For example, see https://arxiv.org/abs/2305.16130, https://arxiv.org/pdf/2303.04245, and the references we mention in Section 4).
>
> Trying to understand this behavior directly in large language models is an admirable goal to strive towards. However, we believe it is too difficult to tackle directly (for now). Previous efforts in this direction have not gotten very far. A useful way for science to proceed in this situation is to focus on first understanding the mysterious behavior in the simplest possible system that exhibits it. Doing this is one of our primary contributions.
>
> In general, one of the main outstanding questions of deep learning theory is: *what is the mechanism and character of representation learning?* Theorists have very few solvable cases, and these analyses tend to rely on unrealistic assumptions on either the data or the learning task (or both). We believe our result is powerful because it answers this question in a relevant natural language task under very few, relatively weak assumptions.
>
> (Also, it’s a bit surprising that we lack a theory of learning dynamics for one of the foundational algorithms of the field! Even if it *weren’t* directly connected to understanding modern models, for posterity’s sake we should try to solve and deeply understand the small handful of truly breakthrough algorithms in the history of neural learning.)
>
> **On our assumptions.** We respectfully disagree that our solution relies on a lot of unverified assumptions. On the contrary, our result relies on rather few assumptions, especially compared to what is par for the course in theory. We tried to emphasize this point throughout the manuscript. For example, our solution does not rely on any distributional assumptions on the data, which is uncommon in theoretical literature. We clearly list the set of approximations and assumptions in Figure 2 – there are only four in total. The first two are approximations to word2vec which are supported by our empirical checks (see Figure 1, Figure 3, the discussion in Appendices A.3, and Figure 5 in Appendix B). The other two are technical conveniences that simplify the analysis; we expect that they may be relaxed (or possibly eliminated completely) with some additional effort, as is suggested by Figure 2.
>
> In particular, we do *not* believe that the similarity between QWEMs and word2vec is sensitive to the choice of dataset or the model size. (Please do point us to where in the paper this was unclear, as we’d like to clarify this in the writing.) We have done some empirical checks confirming this, as we mention in Appendix A; we will include more details in the final manuscript.
>
> As a side note, since our results are data distribution agnostic, we expect them to carry over to a broad class of linearized contrastive algorithm. For example, if your contrastive loss has the form $\\mathcal{L}(W) = \\mathbb{E}\_\\text{positive pairs}[f^{+}(w^T w’)] + \\mathbb{E}\_\\text{negative pairs}[f^{-}(w^T w’)]$, then you can simply take the quadratic Taylor approximation of your $f^{+}(w^Tw’)$ and $f^{-}(w^T w’)$, complete the square, and obtain a target matrix in terms of the positive and negative distributions for your learning problem of interest. In this sense, the idea behind our Theorem 1 is actually very general. See Appendix D.2 for an example using the SimCLR loss.
>
> **Why does gradient descent in word2vec lead to embeddings similar to QWEM rather than directly approximating PMI?** This is a good question, and the answer is a key insight of our result. The crux of the issue is in adequately handling the rank constraint. (In general, rank-constrained optimization is NP-hard, so handling this properly is a subtle business.) Let us compare the two approaches.
> * The idea behind SVD-approximating PMI is to say “Let us first solve the unconstrained minimizer of the loss (i.e., the PMI matrix) and then, at the end, apply the rank constraint by choosing the closest (in Frobenius norm) rank-$d$ matrix.” This fails. The reason is because the loss basin is extremely wide and shallow, so the global unconstrained minimum is actually *very far* from where the model actually ends up via gradient descent in finite time.
> * Our insight is to change the approach: we say “Let us first approximate the loss landscape itself; then, we can account for the rank constraint *throughout the entire optimization process.*” With the quartic approximation of the loss, this turns out to be solvable. As a result, our prediction for what word2vec learns is significantly more accurate.
> We discuss this at the end of Section 2. We will include more details to clarify this point, as it is central to our result.
>
> **Why does decreasing SNR sometimes still yield high performance?** Although the relationship between SNR and task performance is qualitative rather than quantitative, it’s still quite interesting that the relative ordering of analogy categories is preserved. In other words, SNR (which is computed from the corpus statistics and hyperparameters alone) is a useful measure of how difficult a particular class of analogies is to learn. To our knowledge, this is the first time that a measurable *a priori* proxy for analogy difficulty has been proposed.
>
> This still leaves the question of why, within an analogy category, larger dimension yields better performance despite the degradation in SNR. Our leading hypothesis (corroborated by several empirical checks we did) is that this is an ancillary effect of the benchmark using top-1 accuracy as the error metric. As a concrete example, consider the analogy “France : Paris :: Japan : Tokyo”, satisfying japan+(paris-france) ≈ tokyo. As the effective embedding dimension increases, the SNR may decrease; at the same time, all the embeddings become more spaced out (due to the increased available volume in latent space), meaning there are fewer nearby “competitors” for tokyo. In the large $d$ regime, we empirically found that the embeddings are sufficiently spaced, and tokyo is still the nearest embedding to japan+(paris-france) despite an increase in absolute error. Thus, the measured top-1 performance does not degrade with the increased (relative) noise. Note that unintuitive effects associated with using top-k accuracy have been observed in LLMs as well (see https://arxiv.org/abs/2304.15004). We plan to include this discussion in the final version of the manuscript.
>
> Thanks for the thoughtful comments on our manuscript. Let us know if you have further questions.

---

> > ### Comment · Reviewer_TGpe · 2025-08-07
> >
> > Thank you, the authors have addressed my concerns. I will keep the score.

---

### Official Review · Reviewer_sHUx · 2025-07-02

**Clarity:** 4
**Significance:** 2
**Originality:** 3
**Rating:** 4
**Confidence:** 4

**Summary:**

This paper formulates a quadratic model that closely approximates what is learned from Word2Vec using a Taylor approximation of the loss.
They show the resulting embedding learned on Wikipedia closely matches what Word2Vec does, and works almost as well.

**Questions:**

If I wanted to apply this quadratic approximation to other semi-supervised learning tasks (for instance is other domains like genomics or finance), how would I do that?  What could I learn from this to apply to other areas where semi-supervised methods are still being developed (and transformer based methods may be overkill).

**Ethical Concerns:**

["NO or VERY MINOR ethics concerns only"]

**Final Justification:**

The paper provides a new mathematical/optimization perspective on how the classic word embeddings word2vec form.  This would have been very useful and important 10 years ago, when they were SoTA.  I am convinced it is related to modern LLMs via discussion -- however, this analysis is still not of those models, and do not yet apply there.  Its interesting, but not essential.  I'll be fine if accepted and I lean that way, but also fine if not accepted.

**Limitations:**

Yes.

**Quality:**

4

**Strengths And Weaknesses:**

The paper is (for the most part) very well written.  And I found it very easy to read and understand.


Strengths:
  - The word2vec model is an incredibly influential model, so it's a great case study.

  - The proposed model shows to work much better than the Levy+Goldberg 2015 linear approximation model

  - The results on the one data set are quite convincing that it works well.

  - as mentioned, the paper is very clearly written.


Weaknesses:
  - The word2vec model is nice, but is now outdated, as it thoroughly replaced with contextual embeddings (e.g., BERT) and their derivatives.  The analysis also does not seem to extend to other contemporary models like GloVe or FastText.

  - The embedding seems to do slightly worse than word2vec in tests.  It's similar, but not better.

  - I do not see what the use of this analysis is.  It probably would have been very useful in 2016 or so, but I am less sure now.

---

> ### Author Rebuttal · Authors · 2025-07-30
>
> Thanks very much for your review, we appreciate it. We’re glad you found the result clear and convincing. Some comments regarding your concerns and questions:
>
> **What’s the point of this analysis? Isn’t word2vec outdated?** Though the algorithm itself has now been superseded by larger context-aware embeddings, the core mechanistic idea behind word2vec is back in vogue! Many recent papers empirically report linear structure in the latent representations in LLMs – a direct analog of the word2vec embedding geometry. (For example, see https://arxiv.org/abs/2305.16130, https://arxiv.org/pdf/2303.04245, and the references we mention in Section 4).
>
> Trying to understand this behavior directly in large language models is an admirable goal to strive towards. However, we believe it is too difficult to tackle directly (for now). Previous efforts in this direction have not gotten very far. A useful way for science to proceed in this situation is to focus on first understanding the mysterious behavior in the simplest possible system that exhibits it. Doing this is one of our primary contributions.
>
> In general, one of the main outstanding questions of deep learning theory is: *what is the mechanism and character of representation learning?* Theorists have very few solvable cases, and these analyses tend to rely on unrealistic assumptions on either the data or the learning task (or both). We believe our result is powerful because it answers this question in a relevant natural language task under very few, relatively weak assumptions.
>
> (Also, it’s a bit surprising to lack a theory of learning dynamics for one of the foundational algorithms of the field! Even if it *weren’t* directly connected to understanding modern models, for posterity’s sake we should try to solve and deeply understand the small handful of truly breakthrough algorithms in the history of neural learning.)
>
> **What about GloVe and FastText?** We expect the main idea of our proofs to hold for both these models. FastText is essentially the same architecture and loss as word2vec, and its extra engineering tricks will not change the core story much. GloVe is very similar to word2vec; while word2vec uses a contrastive loss, GloVe instead performs *explicit* least-squares factorization. Incorporating something like our Setting 3.1 in the analysis of the GloVe algorithm would yield results exactly analogous to the ones presented here. We plan to include this discussion and some supporting experiments in the final version of the manuscript.
>
> **How can I apply these results to other self-supervised learning tasks?** Since our results are distribution agnostic, they carry over to pretty much any linearized contrastive algorithm. To be specific, if your contrastive loss has the form $\\mathcal{L}(W) = \\mathbb{E}\_\\text{positive pairs}[f^{+}(w^T w’)] + \\mathbb{E}\_\\text{negative pairs}[f^{-}(w^T w’)]$, then you can simply take the quadratic Taylor approximation of your $f^{+}(w^Tw’)$ and $f^{-}(w^T w’)$, complete the square, and obtain a target matrix in terms of the positive and negative distributions for your learning problem of interest. In this sense, the idea behind our Theorem 1 is actually very general. See Appendix D.2 for an example using the SimCLR loss. To get closed-form learning dynamics, you’d need to find an equivalent of our Setting 3.1. This may be difficult depending on the form of your input data; for instance, it may require a whitening data preprocessing step. The idea is to choose these hyperparameters in such a way that the weighted matrix factorization becomes unweighted. For more details, see Proposition 2 and the subsequent discussion.
>
> Let us know if you have further questions.

---

> > ### Comment · Reviewer_sHUx · 2025-08-03
> >
> > I appreciate and found useful to the answers to my second two questions.  This is enough to maintain my score at 4.
> >
> > For the first question, about what does Word2Vec have to do with modern language models -- I found this answer unsatisfying.  I do know that some ideas from analyzing structure from these pre-BERT word embeddings are being found useful.  But this analysis seems very far removed from understanding an intermediate layer of a transformer network.

---

> ### Author Response · Authors · 2025-08-05
> **Re: relation between word2vec and LLMs**
>
> Thank you for your response! For completeness, here we include some follow-up.
>
> We are curious as to which aspect you found unsatisfying. (This feedback would be useful to us when we discuss our result with other colleagues.) It seems to us that there are several "kinds" of analysis one could hope for regarding LLMs:
> 1. Direct analysis of full optimization dynamics under full transformer architecture and under no assumptions on the data distribution. To our knowledge, this is very difficult and the community does not yet know how to do this.
> 2. Analysis of convergence (without characterizing the optimization dynamics) under simplified transformer architecture.
>     * https://arxiv.org/pdf/2308.16898
>     * https://arxiv.org/pdf/2402.05738
> 3. Analysis of some aspects of optimization dynamics under simplified 1-layer transformer architecture and restrictive synthetic data distribution.
>     * https://arxiv.org/pdf/2411.10830
>     * https://arxiv.org/pdf/2305.16380
>     * https://arxiv.org/abs/2410.09605
>     * https://arxiv.org/pdf/2504.12916
> 4. Analysis of some aspects of final learned representations under simplified 1-layer transformer architecture and restrictive synthetic data distribution.
>      * See many works studying algorithms implemented by transformers for ICL
>      * https://arxiv.org/pdf/2403.03867
> 5. Analysis of _full_ optimization dynamics under very simplified attention-like architecture and _arbitrary_ natural data distribution.
>      * Our current work. (See end of this comment for an argument that word2vec dynamics are derivable from linear transformer dynamics.)
>
> Thus, there are many avenues to making progress in understanding representation learning in LLMs. We simply argue that ours is a promising but under-explored one. We note that, to our knowledge, theoretically understanding the intermediate representations of a large language model is still a very open problem.
>
> We emphasize that understanding LLMs is _not_ a primary goal of this paper. We frame our results purely in terms of understanding word2vec. We believe this alone carries sufficient merit. However, we do believe that our results may provide insight in this broader research project of understanding LLMs.
>
> **Heuristic argument that word2vec dynamics are closely related to transformer learning dynamics.**
> Consider a next-token prediction task being learned by a single-layer linear attention transformer with no positional encoding and weight tying.  The feedforward output is
>
> $$
> \mathrm{logits}(X) = W ((X W Q K^\top W^\top X^\top) X W V)^\top
> $$
>
> where $X\in\mathbb{R}^{L\times V}$ is a prompt matrix with one-hot tokens, $W \in\mathbb{R}^{V\times d}$ is the embedding layer (and unembedding, by weight tying), and $Q,K,V$ are the query/key/value weights. The relevant dimensions are $d$ for the embedding dim, $V$ for the vocabulary size, and $L$ for the prompt length.
>
> Let us assume that the Q, K weights are shared (commonly done in practice, see https://arxiv.org/pdf/2001.04451). We will further assume that Q and K are wide and initialized i.i.d. Gaussian with large variance $\sigma^2$. Then we have that $QK^\top \approx \sigma^2 \mathbb{I}\_{d} $ from concentration of Marchenko-Pastur spectrum in the wide limit, and $\frac{d}{dt} (QK^\top) \approx 0$ (which follows from directly computing gradients). Finally, let us assume for simplicity that we fix $V = \mathbb{I}$.
>
> Now things simplify a lot. At initialization we have
>
> $$
> \mathrm{logits}(X) \approx W ((X W W^\top X^\top) X W)^\top = W (W\_{[L]} W^\top\_{[L]} W\_{[L]})^\top
> $$
>
> where $W\_{[L]} \in \mathbb{R}^{L \times d}$ are the embeddings of the prompt words. Focusing on the final-token logits (which predict the next word)
> $$
> \mathrm{finallogits}(X) \approx W W^\top\_{[L]} W\_{[L]} w\_{final}
> $$
>
> Computing the gradient of the logit w.r.t. the $i$th word in the context, we find that
> $$
> \frac{d \mathrm{finallogits}}{d w\_i} \approx  W (w\_i^\top w\_{final}\mathbb{I}\_d + w\_i w\_{final} ^\top)
> $$
>
> Finally, we recall that the standard cross-entropy loss partitions the vocabulary into positive (true next word) and negative (all other words) distributions. Computing the gradient wrt the loss, we find that the embeddings roughly evolve according to the following rule:
> * words within the prompt $w\_i$ evolve to align towards the true next word.
> * Simultaneously, all the $w\_i$ are pushed away from (the softmax-weighted mean of) all other words in the vocabulary.
>
> This qualitatively recovers the word2vec update rule.

---

> > ### Comment · Reviewer_sHUx · 2025-08-06
> >
> > Thanks for the excellent explanation.  As I said, my concern was:
> >
> > > I do know that some ideas from analyzing structure from these pre-BERT word embeddings are being found useful. But this analysis seems very far removed from understanding an intermediate layer of a transformer network.
> >
> > But your reply provides much more literature and the sort of mathematical connection I did not grasp before.  I now see much more how you hope this connects to modern LLMs.

---

### Official Review · Reviewer_7VDM · 2025-07-02

**Clarity:** 3
**Significance:** 4
**Originality:** 3
**Rating:** 5
**Confidence:** 3

**Summary:**

This paper formulates a quadratic approximation (QWEM) to the contrastive objective of word2vec. QWEM's training dynamics can be solved for in closed form when making some extra assumptions on the parameters (e.g., W = W'). The authors demonstrate the trained QWEM solutions empirically well-approximate the trained word2vec vectors, and better than other proposals (Levy et al 2014). A key insight (Result 3) is that the "natural basis" of the gradient flow dynamics are the eigenvectors of the matrix M* (Eq 5). What follows from this result is that the components are learned sequentially, which makes sense in light of early stopping as regularization. A nice follow-up observation is that task vectors (e.g. the "man"-"woman" vector) empirically appear rank 1, supporting the linear representation hypothesis. Overall, QWEMs appear a useful approximation of word2vec, in a way that not only approximates the end solution (as in Levy and Goldberg 2014) but crucially its training dynamics.

**Questions:**

1. Figure 4: (Bottom.) took me a while to parse. For readability I would add ("SNR") to spike strength and ("analogy accuracy") to "the model's ability to use task vectors for analogy completion".

2. l251 State the benchmark in main text

3. Is it enough for the $R^d$ to be rank-1 (line 261)? If the model is good at analogy completion, then shouldn't the task vectors be exactly equal rather than rank-1, where rank-1 instead implies they are the same up to scaling? If I understand correctly, rank-1 is a necessary but not sufficient condition for good analogy completion-- perhaps it'd be useful to comment on this in the manuscript.

4. Figure 1: Missing discussion in the main text (only reference is line 217). For the left plots, I assume each line is one singular value? It seems like in QWEM, the SVs are learned more "sequentially", but each one is learned earlier than in the word2vec plot. In the word2vec plot, it seems there is more "mixed information" being learned earlier, but each SV individually converges later. Does this imply that QWEM promotes sparsity? And is that a good thing?

**Ethical Concerns:**

["NO or VERY MINOR ethics concerns only"]

**Final Justification:**

Overall nice paper! I disagree with other reviewers that word2vec is trivial and that the insights need to have an application for LLMs. Word2vec is a model that stands on its own, and besides, it is good to have some diversity when the field is already inundated by LLMs.

**Limitations:**

Yes

**Quality:**

4

**Strengths And Weaknesses:**

Disclaimer: I read the paper to the best of my ability, but I'm not well-versed in this literature / the methods. I don't envision changing my score or confidence score as I lack the background to judge the contribution.

## Strengths
1. I found this paper highly readable and clear, even not coming from this background. In particular, I liked that the assumptions were very clearly stated (Setting 3.1 and Lemma 3.1), and later justified empirically with closeness to the word2vec solutions.
2. The theory is backed up with experiments that clearly show how rank increases in the representations with QWEM, and how that mirrors the learning dynamics of word2vec.

## Weaknesses
Had some minor clarity issues, see Questions.

---

> ### Author Rebuttal · Authors · 2025-07-30
>
> Thank you very much for your review! We’re glad you found the paper readable. Thank you for your suggestions regarding clarity – they are very useful to us and we’ll make the requisite changes. Some comments follow regarding your other questions.
>
> **Does early mixing and delayed convergence imply that QWEM promotes sparsity?** Both QWEMs and word2vec exhibit a low-rank bias in their learning dynamics, although the effect is more pronounced in QWEMs, as you observe. We don’t believe that this low-rank bias is necessarily good or bad; that depends on the downstream task of interest. Our theory is simply descriptive of the learning dynamics.
>
> The reason that clear stepwise learning occurs in QWEMs is that our Setting 3.1 turns the weighted factorization problem into an unweighted factorization – as a consequence, the singular value dynamics become decoupled (i.e., “untangled”). Without our Setting 3.1, there is “mixing” between the singular directions (as you observe), even when the corresponding singular values have grown to be $O(1)$ in magnitude. Exactly characterizing this mixing is NP hard (see Proposition 2 and the subsequent discussion). The somewhat surprising fact is that word2vec is “close enough” to the unweighted factorization – though the singular value dynamics do exhibit some early mixing, the overall learning dynamics are well-described by the simplified setting. This agreement is partly because the hyperparameters chosen in word2vec coarsely approximate our Setting 3.1 (see Appendix A.3).
>
> As for the slightly delayed convergence, we don’t think this is an important effect. The two algorithms optimize different objectives, so the magnitudes of the gradients are not exactly the same; one would not expect their learning timescales to match exactly. Our theory predicts the convergence time with relatively small error – this is already much better than previous results, which only give the scaling behavior (or loose bounds) for the convergence time, rather than a direct and accurate prediction.
>
> **Shouldn't the task vectors be exactly equal rather than rank-1?** Strictly speaking, for exact analogy recovery, you’re right that rank-1 task vectors are necessary but not sufficient. In practice though, when performance is measured with top-1 accuracy, the length of the task vectors is rather unimportant in the high-dimensional regime. The angular information is far more relevant. (One can intuitively see this by considering the toy setting of Gaussian random vectors in the high-dimensional limit – their lengths concentrate and only angles matter.) This is especially true since the embeddings are normalized to be unit norm before evaluating the analogy task. We discuss this subtlety in Appendices A.4 and A.6. We will clarify this in the main text.
>
> Let us know if you have further questions!

---

> > ### Comment · Reviewer_7VDM · 2025-08-01
> >
> > Thanks for answering my questions, esp. regarding the rank-1 task vectors. I will maintain my score. Great work!

---

### Official Review · Reviewer_WyQ8 · 2025-07-03

**Clarity:** 4
**Significance:** 4
**Originality:** 4
**Rating:** 5
**Confidence:** 2

**Summary:**

The authors present an analysis of the word2vec skip-gram negative sampling algorithm. They derive an exact minimizer of a Taylor approximation to the skip-gram objective (QWEM), and show that the optimal solution of QWEM is related to the PMI matrix describing word-word co-occurrences. Stochastic gradient optimization of QWEM yields a sequence of discrete rank-incrementing learning steps that correspond to substantial decreases in loss; the semantics of the additional dimensions added at a step $t$ correspond to dimensions added in a similar learning step in word2vec. QWEM can be optimized in either this stochastic way or solved for in closed-form; the resulting word vectors perform similarly to those from word2vec. A final section studies the geometry underlying linear vector analogy in the QWEM embedding space.

**Questions:**

- How do you account for the dissociation between SNR and accuracy in higher dimensions (the right halves of the bottom-right plots in Figure 4; compare e.g. the blue curves, where the SNR curve peaks and then decreases while the accuracy curve appears to monotonically increase)? If it is the case that this lone "spike" eigenvalue is the task-relevant direction, why is it the case that decreasing SNR at higher dimensions still yields higher performance?
- Why do we not see the same clear discrete rank-incremental performance in the word2vec loss function?

**Ethical Concerns:**

["NO or VERY MINOR ethics concerns only"]

**Final Justification:**

Some other reviewers questioned whether this theoretical study of word2vec is useful in an age when word2vec has been replaced by LLMs as the primary object of study. I agree with the authors' rebuttal here: word2vec still establishes a relatively simple setting for preliminary theoretical analysis, and there is good chance (given some convergent findings about linear geometry in both spaces) that some of the insights could generalize to other more modern word embedding methods.

I will keep my score.

**Limitations:**

Yes

**Quality:**

4

**Strengths And Weaknesses:**

This is an extremely clear and interesting paper studying what is now a fundamental algorithm of the field, combining theoretical work with thorough empirical validation. This is an important result and could serve as the foundation both for further analysis of word2vec and for intuition about the function of more complex models.

The relationship between Section 4 and the rest of the paper is unclear to me — it seems like this is mostly a free-standing study, in principle independent of the theory motivated in earlier sections. Perhaps the authors want to link the claimed training dynamics (rank-incrementing steps of training with interpretable semantics) to the notion of task vectors operant in analogy evaluations — but from what I can see in fig. 4 top right, these task vectors don't have a strong one-to-one alignment with the model eigenfeatures; instead they combine many of those dimensions.

---

> ### Author Rebuttal · Authors · 2025-07-30
>
> Thank you very much for your review! We’re glad you found the paper clear and interesting. Some comments follow.
>
> **Why does decreasing SNR sometimes still yield high performance?** This is a very interesting question and we spent some time exploring this. Our leading hypothesis (corroborated by several empirical checks we did) is that this is an ancillary effect of the benchmark using top-1 accuracy as the error metric. As a concrete example, consider the analogy “France : Paris :: Japan : Tokyo”, satisfying japan+(paris-france) ≈ tokyo. As the effective embedding dimension increases, the SNR may decrease; at the same time, all the embeddings become more spaced out (due to the increased available volume in latent space), meaning there are fewer nearby “competitors” for tokyo. In the large $d$ regime, we empirically found that the embeddings are sufficiently spaced, and tokyo is still the nearest embedding to japan+(paris-france) despite an increase in absolute error. Thus, the measured top-1 performance does not degrade with the increased (relative) noise. Note that unintuitive effects associated with using top-k accuracy have been observed in LLMs as well (see https://arxiv.org/abs/2304.15004). We plan to include this discussion in the final version of the manuscript.
>
> **Where is the clear, discrete, rank-incrementing behavior in bona fide word2vec?** You correctly observe that the rank-incrementing steps in word2vec do not appear clearly separated (especially early in training). The reason that clear stepwise learning occurs in QWEMs is that our Setting 3.1 turns the weighted factorization problem into an unweighted factorization – as a consequence, the singular value dynamics become decoupled (i.e., “untangled”). Without our Setting 3.1, there is “mixing” between the singular directions, even when the corresponding singular values have grown to be $O(1)$ in magnitude. Exactly characterizing this mixing is NP hard (see Proposition 2 and the subsequent discussion). The somewhat surprising fact is that word2vec is “close enough” to the unweighted factorization – though the singular value dynamics do exhibit some mixing (as you observe), the overall learning dynamics are well-described by the simplified setting. This agreement is partly because the hyperparameters chosen in word2vec coarsely approximate our Setting 3.1 (see Appendix A.3).
>
> **Regarding the link between Section 4 and the theoretical results.** Though the empirical results in Section 4 do not rely on the correctness of the preceding theory, the two are related. Our first result (left and top right of Fig. 4) suggests that it may be possible to obtain a *sparse* effective theory of semantic linear representations by studying their decomposition in terms of the semantic basis identified by our theory. Our second result (bottom right) identifies a potential benefit of early stopping in large models – it may prevent the realization of noisy semantic directions in the latents. This insight follows from the duality between model size and training time identified by our theory (i.e., the low-rank bias). In general, all the empirics in Section 4 were relatively easy to iterate on because they only involved factorizing a large matrix once, rather than re-training many word embedding models. This computational saving follows directly from the theory (we discuss it in Section 3.3).
>
> Let us know if you have further questions!

---

> > ### Comment · Area_Chair_MDxD · 2025-08-07
> > **Please Engage in Author Response Discussion**
> >
> > Dear Reviewer WyQ8,
> >
> > We encourage you to review the authors’ rebuttals and see how they’ve addressed your comments. If you’ve already done so, thank you! Kindly confirm your engagement by reacting in this thread or leaving a brief comment.
> >
> > Your participation helps ensure a fair and thoughtful review process.
> >
> > Best regards,
> > AC

---

> > ### Comment · Reviewer_WyQ8 · 2025-08-07
> >
> > Hi authors, thank you very much for the clarifications. I agree with your point made elsewhere in the rebuttal that this is a theoretically important finding, even though word2vec itself is no longer at the forefront. I will keep my score.

---

### Decision · Program_Chairs · 2025-09-17

**Decision:**

Accept (poster)

**Comment:**

This paper analyzes word2vec by deriving a quartic approximation, which exhibits training dynamics and downstream performance closely aligned with the original model.

The reviewers recognized the solid technical contributions and found the analysis interesting and valuable, as reflected in their positive evaluations. No major concerns were raised, and the author rebuttal addressed the reviewers’ questions satisfactorily.

As noted by the reviewers, word2vec has become a classic approach for obtaining word embeddings. While the work may not appear super exciting, it nonetheless provides a rigorous analysis. Given its solid contribution to word embedding, this paper represents an important and worthwhile addition to NeurIPS.